# A Black-box Approach for Non-stationary Multi-agent Reinforcement Learning

**Haozhe Jiang**[1]    **Qiwen Cui**[2]    **Zhihan Xiong**[2]    **Maryam Fazel**[2]    **Simon S. Du**[2]

[1] Institute for Interdisciplinary Information Sciences, Tsinghua University
[2] University of Washington

## Abstract

We investigate learning the equilibria in non-stationary multi-agent systems and address the challenges that differentiate multi-agent learning from single-agent learning. Specifically, we focus on games with bandit feedback, where testing an equilibrium can result in substantial regret even when the gap to be tested is small, and the existence of multiple optimal solutions (equilibria) in stationary games poses extra challenges. To overcome these obstacles, we propose a versatile black-box approach applicable to a broad spectrum of problems, such as general-sum games, potential games, and Markov games, when equipped with appropriate learning and testing oracles for stationary environments. Our algorithms can achieve $\widetilde{O}\left(\Delta^{1/4}T^{3/4}\right)$ regret when the degree of nonstationarity, as measured by total variation $\Delta$, is known, and $\widetilde{O}\left(\Delta^{1/5}T^{4/5}\right)$ regret when $\Delta$ is unknown, where $T$ is the number of rounds. Meanwhile, our algorithm inherits the favorable dependence on number of agents from the oracles. As a side contribution that may be independent of interest, we show how to test for various types of equilibria by a black-box reduction to single-agent learning, which includes Nash equilibria, correlated equilibria, and coarse correlated equilibria.

## 1 Introduction

Multi-agent reinforcement learning (MARL) studies the interactions of multiple agents in an unknown environment with the aim of maximizing their long-term returns (Zhang et al., 2021). This field has applications in diverse areas such as computer games (Vinyals et al., 2019), robotics (de Witt et al., 2020), and smart manufacturing (Kim et al., 2020). Although various algorithms have been developed for MARL, it is typically assumed that the underlying repeated game is stationary throughout the entire learning process. However, this assumption often fails to represent real-world scenarios where the environment is evolving throughout the learning process.

The task of learning within a non-stationary multi-agent system, while crucial, poses additional challenges when attempts are made to generalize non-stationary single-agent reinforcement learning (RL), especially for the bandit feedback case where minimal information is revealed to the agents (Anagnostides et al., 2023). In addition, the various multi-agent settings, such as zero-sum, potential, and general-sum games, along with normal-form and extensive-form games, and fully observable or partially observable Markov games, further complicate the design of specialized algorithms.

In this work, we take the first step towards understanding non-stationary MARL with bandit feedback. First, we point out several challenges that differentiate non-stationary MARL from non-stationary single-agent RL, and bandit feedback from full-information feedback. Subsequently, we propose black-box algorithms with sub-linear dynamic regret in arbitrary non-stationary games, provided there is access to learning algorithms in the corresponding (near-)stationary environment. This versatile approach allows us to leverage existing algorithms for various stationary games, while facilitating seamless adaptation to future algorithms that may offer improved guarantees.

### 1.1 Main Contributions and Novelty

**1. Identifying challenges in non-stationary games with bandit feedback (Section 3).** First, we point out that bandit feedback is incompatible with online-learning based algorithms as the gradient

of reward is hard to estimate. Then, we show that bandit feedback complicates the application of test-based algorithms as testing an arbitrary small gap can incur $O(1)$ regret each term. Non-uniqueness of equilibria makes replay-based test difficult as well. Additionally, we point out that it is non-trivial to generalize an algorithm for non-stationary Markov games to a parameter-free version since the objective for games is very different from that of multi-armed bandits.

**2. Generic black-box approach for non-stationary games.** Our approach is a black-box reduction that can transform any base algorithm designed for (near-)stationary games into an algorithm capable of learning in a non-stationary environment. This approach inherits favorable properties of the base algorithm, like breaking the curse of multi-agents, and directly adapts to future algorithmic advances.

**3. Restart-based algorithm when non-stationarity budget is known (Section 4).** When we know a bound on the degree of non-stationarity, often measured by number of switches or total variation (which from here on, we refer to as the "nonstationarity budget"), we design a simple restart-based algorithm achieving sublinear dynamic equilibrium regret of $\widetilde{O}(L^{1/4}T^{3/4})$ or $\widetilde{O}(\Delta^{1/4}T^{3/4})$, where $L$ is the switching number and $\Delta$ is the total variation non-stationarity budget. In words, this result implies that all the players follow a near-equilibrium strategy in most episodes.

**4. Multi-scale testing algorithm when non-stationarity budget is unknown (Section 5).** We also propose a multi-scale testing algorithm to optimize the regret when the non-stationarity budget is unknown, which can adaptively avoid the strategy deviating from equilibrium for too many rounds. The algorithm achieves the same $\widetilde{O}(L^{1/4}T^{3/4})$ regret for unknown switching number $L$, and a marginally higher $\widetilde{O}(\Delta^{1/5}T^{4/5})$ regret for unknown total variation budget $\Delta$. The testing algorithms are newly designed and the scheduling is specially designed for the PAC assumptions, which is different from that in Wei & Luo (2021) where regret assumptions are made.

While the ultimate goal is to design no-regret algorithms for each agent, i.e. achieving no-regret no matter what policy other players adopt (like Panageas et al. (2023)), our setting is already applicable in various real-world cases even without yet achieving this desired property, this is discussed with a concrete example below. We leave the problem of finding no-regret algorithms for each individual for future work.

**Example (traffic routing with navigation).** In traffic routing using navigation applications (Guo et al. (2023)), being able to track Nash Equilibrium is advantageous. Assume all drivers use the same navigation application which runs our algorithm. It is reasonable to assume that drivers adhere to the application's suggestions. After following the route recommended by the application, the drivers all find that their routes are not improvable because all drivers are committing to the equilibrium; this makes drivers satisfied with the algorithm's recommendation.

## 1.2 RELATED WORK

Due to the space limit, we defer the more comprehensive related work to the appendix.

**(Stationary) Multi-agent reinforcement learning.** Numerous works have been devoted to learning equilibria in (stationary) multi-agent systems, including zero-sum Markov games (Bai et al., 2020), general-sum Markov games (Jin et al., 2021), Markov potential games (Cui et al., 2023), congestion games (Cui et al., 2022), extensive-form games (Kozuno et al., 2021), and partially observable Markov games (Liu et al., 2022). These works aim to learn equilibria with bandit feedback efficiently, measured by either regret or sample complexity.

**Non-stationary (single-agent) reinforcement learning.** The study of non-stationary reinforcement learning originated from non-stationary bandits (Auer et al., 2002). Auer et al. (2019) and Chen et al. (2019) first achieve near-optimal dynamic regret without knowing the non-stationary budget for bandits. The most relevant work is Wei & Luo (2021), which also proposes a black-box approach with multi-scale testing and achieves optimal regret in various single-agent settings.

## 2 PRELIMINARIES

We consider the multi-agent general-sum Markov games framework, which covers a wide range of problems, and is described by the tuple $\mathcal{M} = (\mathcal{S}, \mathcal{A} = \mathcal{A}_1 \times \cdots \times \mathcal{A}_m, H, \mathbb{P}, \{r_i\}_{i=1}^m)$, where $\mathcal{S}$ is the state space with cardinality $S$, $m$ is the number of the players, $\mathcal{A}_i$ is the action space for player $i$

with cardinality $A_i$, $H$ is the length of the horizon, $\mathbb{P} = \{\mathbb{P}_h\}_{h=1}^H$ is the collection of the transition kernels. $P_h : \mathcal{S} \times \mathcal{A} \rightarrow \Delta(\mathcal{S})$ and $\mathbb{P}_h(\cdot \mid s, \mathbf{a})$ is the next state distribution given the current state $s$ and joint action $\mathbf{a} = (a_1, \cdots, a_m)$ at step $h$. $r_i = \{r_{h,i}\}_{h=1}^H$ is the collection of random reward functions for player $i$; $r_{h,i} : \mathcal{S} \times \mathcal{A} \rightarrow [0, 1]$, and its mean is denoted as $R_{h,i}$. At the beginning of each episode, the players start at a fixed initial state $s_1$.[1] At each step $h \in [H]$, each player observes the current state $s_h$ and chooses action $a_{h,i}$ simultaneously. Then player $i \in [m]$ will receive her own reward realization $\widetilde{r}_{h,i} \sim r_{h,i}(s_h, \mathbf{a}_h)$ where $\mathbf{a}_h = (a_{h,1}, \cdots, a_{h,m})$ and the state will transition according to $s_{h+1} \sim \mathbb{P}_h(\cdot \mid s_h, \mathbf{a}_h)$. The game will terminate when $h = H + 1$. We consider the bandit feedback setting where only the reward of the chosen action is revealed to the player.

Here we discuss the generality of Markov games. When the horizon $H = 1$, multi-player general-sum Markov games reduce to multi-player general-sum matrix games, which include zero-sum games, potential games, congestion games, etc. (Nisan et al., 2007). If we pose different assumptions on the Markov game structure, we can obtain zero-sum Markov games (Bai et al., 2020), Markov potential games (Leonardos et al., 2021), extensive-form games (Kozuno et al., 2021). If the state $s_h$ is not directly observable, the Markov games are modeled by partially observable Markov games (Liu et al., 2022). A detailed section on preliminaries for different games is deferred to the appendix.

**Policy.** A Markov joint policy is defined as $\pi = \{\pi_h\}_{h=1}^H$ where $\pi_h : \mathcal{S} \rightarrow \Delta(\mathcal{A})$ is the policy at step $h$. We will use $\pi_{-i}$ to denote that all players except for player $i$ are following policy $\pi$. A special case of Markov joint policy is Markov product policy, which satisfies that there exist policies $\{\pi_i\}_{i=1}^m$ such that for all $h \in [H]$ and $(s, \mathbf{a}) \in \mathcal{S} \times \mathcal{A}$, we have $\pi_h(\mathbf{a} \mid s) = \prod_{i=1}^m \pi_{h,i}(a_i \mid s)$, where $\pi_i = \{\pi_{h,i}\}_{h=1}^H$ is the collection of Markov policies $\pi_{h,i} : \mathcal{S} \rightarrow \Delta(\mathcal{A}_i)$ for player $i$. In words, a Markov product policy can be factorized into individual policies such that they are uncorrelated.

**Value function.** Given a Markov game $M \in \mathcal{M}$ and a policy $\pi$, the value function for player $i$ is defined as $V_i^M(\pi) := \mathbb{E}_\pi \left[ \sum_{h=1}^H r_{h,i}(s_h, \mathbf{a}_h) \Big| M \right]$, where the expectation is taken over the randomness in both the policy and the environment.

**Best response and strategy modification.** Given a policy $\pi$ and Markov game $M$, the best response value for player $i$ is $V_i^M(\dagger, \pi_{-i}) := \max_{\pi_i' \in \Pi_i} V_i^M(\pi_i', \pi_{-i})$, which is the maximum achievable expected return for player $i$ if all the other players follow $\pi_{-i}$. Equivalently, best response is the optimal policy in the induced Markov decision process (MDP), i.e., Markov game with one player.

A strategy modification $\psi_i = \{\psi_{h,i}\}_{h=1}^H$ is a collection of mappings $\psi_{h,i} : \mathcal{S} \times \mathcal{A}_i \rightarrow \mathcal{A}_i$ that maps the joint state-action space to the action space.[2] For policy $\pi$, $\psi_i \diamond \pi$ is the modified policy such that

$$(\psi_i \diamond \pi)_h(\mathbf{a} \mid s) = \sum_{\mathbf{a}' : \psi_{h,i}(a_i'|s)=a_i, \mathbf{a}'_{-i}=\mathbf{a}_{-i}} \pi_h(\mathbf{a}' \mid s).$$

In other words, $\psi_i \diamond \pi$ is a policy such that if $\pi$ assigns each player $j$ a random action $a_j$ at state $s$ and step $h$, then $\psi_i \diamond \pi$ assigns action $\psi_{h,i}(a_i \mid s)$ to player $i$ while all the other players are following the action assigned by policy. We will use $\Psi_i$ to denote all the possible strategy modifications for player $i$. As $\Psi_i$ contains all the constant strategy modifications, we have

$$\max_{\psi_i \in \Psi_i} V_i^M(\psi_i \diamond \pi) \geq \max_{\pi_i'} V_i^M(\pi_i', \pi_{-i}) = V_i^M(\dagger, \pi_{-i}),$$

which means that the best strategy modification is always no worse than the best response.

**Notions of equilibria.**

**Definition 1.** *For Markov game $M$, policy $\pi$ is an $\epsilon$-approximate Nash equilibrium (NE) if it is a product policy and*

$$\mathrm{NEGap}^M(\pi) = \max_{i \in [m]} \left( V_i^M(\dagger, \pi_{-i}) - V_i^M(\pi) \right) \leq \epsilon.$$

Learning Nash equilibrium (NE) is neither computationally nor statistically efficient for general-sum normal-form games (Chen et al., 2009), while it is tractable for games with special structures, such as potential games (Monderer & Shapley, 1996) and two-player zero-sum games (Adler, 2013).

---

[1]It is straightforward to generalize to stochastic initial state by adding a dummy state $s_0$ that transition to the random initial state.

[2]We only consider deterministic strategy modification as the optimal strategy modification can always be deterministic (Jin et al., 2021).

**Definition 2.** *For Markov game $M$, policy $\pi$ is an $\epsilon$-approximate coarse correlated equilibrium (CCE) if*

$$\mathrm{CCEGap}^M(\pi) = \max_{i \in [m]} \left( V_i^M(\dagger, \pi_{-i}) - V_i^M(\pi) \right) \le \epsilon.$$

The only difference between CCE and NE is that CCE is not required to be a product policy. This relaxation allows tractable algorithms for learning CCE.

**Definition 3.** *For Markov game $M$, policy $\pi$ is an $\epsilon$-approximate correlated equilibrium (CE) if*

$$\mathrm{CEGap}^M(\pi) = \max_{i \in [m]} \left( \max_{\psi_i \in \Psi_i} V_i^M(\psi_i \diamond \pi) - V_i^M(\pi) \right) \le \epsilon.$$

Correlated equilibrium generalizes the best response used in CCE to best strategy modification. It is known that each NE is a CE and each CE is a CCE. For conciseness, we use $\epsilon$-EQ to denote $\epsilon$-approximate NE/CE/CCE.

**Non-stationarity measure.** Here we formalize the non-stationary Markov game. There are $T$ total episodes and at each episode $t$, the players are following some policy $\pi^t$ an unknown Markov game $M^t$. The non-stationarity degree of the environment is measured by the cumulative difference between two consecutive models, defined as follows.

**Definition 4.** *The non-stationarity degree of Markov games $(M^1, M^2, \cdots, M^T)$ is measured by total variation $\Delta$ or number of switches $L$, which are respectively defined as*

$$\Delta = \sum_{t=1}^{T-1} \left\| M^{t+1} - M^t \right\|_1, \quad L = \sum_{t=1}^{T-1} \mathbb{1}[M^t \ne M^{t+1}].$$

*Here, the total variation distance between two Markov games is defined as*

$$\left\| M - M' \right\|_1 := \sum_{h=1}^{H} \left( \max_{s \in \mathcal{S}, \mathbf{a} \in \mathcal{A}} \left\| \mathbb{P}_h^M(\cdot | s, \mathbf{a}) - \mathbb{P}_h^{M'}(\cdot | s, \mathbf{a}) \right\|_1 + \max_{s \in \mathcal{S}, \mathbf{a} \in \mathcal{A}, i \in [m]} \left| R_{h,i}^M(s, \mathbf{a}) - R_{h,i}^{M'}(s, \mathbf{a}) \right| \right)$$

*We also define*

$$\Delta_{[t_1, t_2]} = \sum_{t=t_1}^{t_2-1} \left\| M^{t+1} - M^t \right\|_1, \quad L_{[t_1, t_2]} = \sum_{t=t_1}^{t_2-1} \mathbb{1}[M^t \ne M^{t+1}].$$

**Dynamic regret.** We generalize the standard dynamic regret in non-stationary single-agent RL to non-stationary MARL.

**Definition 5.** *The dynamic equilibrium regret is defined as*

$$\mathrm{Regret}(T) = \sum_{t=1}^{T} \mathrm{Gap}^{M^t}(\pi^t),$$

*where $\mathrm{Gap}(\cdot)$ can be* NEGap, CCEGap *or* CEGap.

A small dynamic regret implies that for most episodes $t \in [T]$, the policy $\pi^t$ is an approximate equilibrium for model $M^t$. The same dynamic regret is used in Anagnostides et al. (2023) for matrix games. In the literature, Cardoso et al. (2019) and Zhang et al. (2022) propose NE-regret and dynamic NE-regret for two-player zero-sum games where the comparator is the best NE value in hindsight and the best dynamic NE value. However, these regret notions cannot be generalized to general-sum games as the NE/CE/CCE values become non-unique. Zhang et al. (2022) also considers duality gap as a performance measure, which coincides with our dynamic regret where $\mathrm{Gap}$ is NEGap.

**Base algorithms.** Our algorithm will use black-box oracles that can learn and test equilibria in a (near-)stationary environment. Details of the base algorithms are shown in Appendix.

**Assumption 1.** *(PAC guarantee for learning equilibrium) We assume that we have access to an oracle* LEARN_EQ *such that with probability $1 - \delta$, in an environment with non-stationarity $\Delta$ as defined in Definition 4, it can output an $(\epsilon + c_1^\Delta \Delta)$-EQ of a game with at most $C_1(\epsilon, \delta)$ samples.*

Table 1: $A, B$ are the size of action spaces for two-player zero-sum games. $A_i$ is the number of actions for player $i$. $A_{\max} = \max_{j \in [m]} A_j$. $S$ is the size of the state space, $H$ is the horizon of the Markov games and $T$ is the number of episodes. The second and third column is the sample complexity for learning and testing an equilibrium in a stationary game. The last column shows the regret bounds for Algorithm 1. References and algorithmic details can be found in Appendix E.

| Types of Games | LEARN_EQ | TEST_EQ | Dynamic Regret |
|---|---|---|---|
| Zero-sum (NE) | $(A+B)\epsilon^{-2}$ | $(A+B)\epsilon^{-2}$ | $((A+B)\Delta)^{1/4}T^{3/4}$ |
| General-sum (CCE) | $A_{\max}\epsilon^{-2}$ | $mA_{\max}\epsilon^{-2}$ | $(A_{\max}\Delta)^{1/4}T^{3/4}$ |
| General-sum (CE) | $A_{\max}^2\epsilon^{-2}$ | $mA_{\max}^2\epsilon^{-2}$ | $(A_{\max}^2\Delta)^{1/4}T^{3/4}$ |
| Potential (NE) | $m^2 A_{\max}\epsilon^{-3}$ | $mA_{\max}\epsilon^{-2}$ | $(m^2 A_{\max}\Delta)^{1/5}T^{4/5}$ |
| Congestion (NE) | $m^2 F^3\epsilon^{-2}$ | $mF^2\epsilon^{-2}$ | $(m^3 F^4\Delta)^{1/4}T^{3/4}$ |
| Zero-sum Markov (NE) | $H^5 S(A+B)\epsilon^{-2}$ | $H^3 S(A+B)\epsilon^{-2}$ | $(H^7 S(A+B)\Delta)^{1/4}T^{3/4}$ |
| General-sum Markov (CCE) | $H^6 S^2 A_{\max}\epsilon^{-2}$ | $mH^3 SA_{\max}\epsilon^{-2}$ | $(H^7 S^3 A_{\max}\Delta)^{1/4}T^{3/4}$ |
| General-sum Markov (CE) | $H^6 S^2 A_{\max}^2\epsilon^{-2}$ | $mH^3 SA_{\max}^2\epsilon^{-2}$ | $(H^7 S^3 A_{\max}^2\Delta)^{1/4}T^{3/4}$ |
| Markov Potential (NE) | $m^2 H^4 SA_{\max}\epsilon^{-3}$ | $mH^3 SA_{\max}\epsilon^{-2}$ | $(m^2 H^6 SA_{\max}\Delta)^{1/5}T^{4/5}$ |

**Assumption 2.** *(PAC guarantee for testing equilibrium) We assume that we have access to an oracle* TEST_EQ *such that given a policy $\pi$, with probability $1 - \delta$, in an environment with non-stationarity $\Delta$ as defined in Definition 4, it outputs False when $\pi$ is not a $(2\epsilon + c_2^\Delta \Delta)$-EQ for all $t = 1, \ldots, C_2(\epsilon, \delta)$ and outputs True when $\pi$ is an $(\epsilon - c_2^\Delta \Delta)$-EQ for all $t = 1, \ldots, C_2(\epsilon, \delta)$.*

There exist various algorithms (see Table 1) providing PAC guarantees for learning equilibrium in stationary games, which satisfies Assumption 1 when non-stationarity degree $\Delta = 0$. We will show that most of these algorithms enjoy an additive error w.r.t. non-stationarity degree $\Delta$ in Appendix E and discuss how to construct oracles satisfying Assumption 2 in Section 5.1. For simplicity, we will omit $\delta$ in $C_1(\epsilon, \delta)$ and $C_2(\epsilon, \delta)$ as they only have poly-logarithmic dependence on $\delta$ for all the oracle realizations in this work. Furthermore, since the dependence of $C_1(\epsilon), C_2(\epsilon)$ on $\epsilon$ are all polynomial, we denote $C_1(\epsilon) = c_1\epsilon^\alpha, C_2(\epsilon) = c_2\epsilon^{-2}$. Here $c_1, c_2$ does not depend on $\epsilon$ and $\alpha$ is a constant depending on the oracle algorithm. In Table 1, $\alpha = -2$ or $\alpha = -3$, where $\alpha$ is the exponent in $C_1(\epsilon)$.

## 3 CHALLENGES IN NON-STATIONARY GAMES

In this section, we discuss the major difficulties generalizing single-agent non-stationary algorithms to non-stationary Markov games. There are two major lines of work in the single-agent setting. The first line of work uses online learning techniques to tackle non-stationarity. There exist works generalizing online learning algorithms to the multi-agent setting. However most of them apply only to the full-information setting. In the bandit feedback setting, it is hard to estimate the gradient of the objective function. The other line of work uses explicit tests to determine notable changes of the environment and restart the whole algorithm accordingly. This paper also adopts this paradigm.

The first type of test is to play a sub-optimal action $a$ consecutively to determine whether it has become optimal (Auer et al., 2019; Chen et al., 2019). For simplicity, let us think of learning NE in the environment with abrupt changes (switching number as the non-stationary measure). In order to assure $a$ has not become a new optimal action, one needs to spend $1/D^2$ steps to play $a$ and secure its value up to confidence bound $D$ where $D$ is the suboptimality.[3] The regret incurred in this testing process is $D \cdot 1/D^2 = 1/D$. In the multi-agent setting, if one wants to repeat the process by testing $(a_i', a_{-i})$ to assure $\boldsymbol{a}$ is a NE, the timesteps needed is still $1/D^2$ where $D$ is the empirical reward difference of $(a_i', a_{-i})$ and $\boldsymbol{a}$. However, the gap of $(a_i', a_{-i})$ depends on its own unilateral deviations, which can be $O(1)$ in general. Hence the regret incurred can be $1/D^2$, sabotaging the test process (example in Figure 1) and greatly increase the regret.

The second type of test restarts the learning algorithm for a small amount of time and checks for abnormality in the replay (Wei & Luo, 2021). In the multi-agent setting, since equilibrium is not unique in all games, different runs of the same algorithm can converge to different equilibria even in a stationary environment. Hence test of this type fails to detect abnormality in the base algorithm.

---

[3] $1/D^2$ is the statistical lower bound.

Another method worth mentioning was invented in Garivier & Moulines (2011). This method proposes to forget old history through putting a discount weight on old feedback or imposing a sliding window based on which we calculate the empirical estimate of value of actions. There is no obvious obstacle in generalizing it to the multi-agent setting but it is hard to derive a parameter-free version. Cheung et al. (2020) uses the Bandit-Over-RL technique to get a parameter-free version for the single-agent setting based on the sliding-window idea. However, the Bandit-Over-RL technique does not generalize to the multi-agent setting as the learning objective is totally different. A more detailed version of the challanges mentioned is presented in the Appendix B.

|       | $a$ | $b$         |
|-------|-----|-------------|
| $a$   | 1   | 0           |
| $b$   | 0   | $\varepsilon$ |

Figure 1: Consider a two-player cooperative game. Both players have access to action space $\{a, b\}$ and the corresponding rewards are shown in the picture. Assume we have found NE $(b, b)$. If we want to make sure $(a, b)$ has not become a best response for player 1, we have to play $(a, b)$ for $1/\varepsilon^2$ times. However the regret of $(a, b)$ is 1, so this process induces $1/\varepsilon^2$ regret.

## 4 Warm-Up: Known Non-Stationary Budget

We first present an algorithm for MARL against non-stationary environments with known non-stationarity budget to serve as a starting point.

---

**Algorithm 1** Restarted Explore-then-Commit for Non-stationary MARL

---

1: **Input:** number of episodes $T$; non-stationarity budget $\Delta$; confidence level $\delta$; parameter $T_1$
2: **while** episode $T$ is not reached **do**
3:      Run Learn_EQ with accuracy $\epsilon$ and confidence level $\delta$, and receive the output $\pi$.
4:      Execute $\pi$ for $T_1$ episodes.

---

Initially, the algorithm starts a Learn_EQ algorithm, intending to learn an $\epsilon$-EQ policy $\pi$. After that, it commits to $\pi$ for $T_1$ episodes. Subsequently, the algorithm repeats this learn-then-commit pattern until the end. The restart mechanism guarantees that the non-stationarity in the environment can at most affect $T_1$ episodes. By carefully tuning $T_1$, we can achieve a sublinear regret. This algorithm admits a performance guarantee as follows.

**Proposition 1.** *With probability* $1 - T\delta$*, the regret of Algorithm 1 satisfies*

$$\text{Regret}(T) \leq \frac{4TC_1(\epsilon)}{T_1} + T\epsilon + 2\max\left\{c_1^\Delta, H\right\}T_1\Delta.$$

*Remark* 1. Let us look at the meaning of each term in this bound. The first term comes from all Learn_EQ. The second and third terms come from committing to the learned policy.

**Corollary 1.** *With probability* $1 - T\delta$*, the regret of Algorithm 1 satisfies*

$$\text{Regret}(T) \leq \begin{cases} 13\left(\Delta c_1\max\left\{c_1^\Delta, H\right\}\right)^{1/4}T^{3/4}, & \alpha = -2, \\ 13\left(\Delta c_1\max\left\{c_1^\Delta, H\right\}\right)^{1/5}T^{4/5}, & \alpha = -3, \end{cases}$$

*by setting*

$$T_1 = \left\lceil \sqrt{\frac{TC_1(\epsilon)}{\max\left\{c_1^\Delta, H\right\}\Delta}} \right\rceil, \quad \epsilon = \begin{cases} \left(\Delta c_1\max\left\{c_1^\Delta, H\right\}/T\right)^{1/4}, & \alpha = -2, \\ \left(\Delta c_1\max\left\{c_1^\Delta, H\right\}/T\right)^{1/5}, & \alpha = -3. \end{cases}$$

*Example* 1. As a concrete example, for learning CCE in general-sum Markov games, Algorithm 1 achieves $O(A_{\max}^{1/4}\Delta^{1/4}T^{3/4})$ regret. We can see that this algorithm breaks the curse of multi-agents (dependence on the number of players) which is a nice property inherited from the base algorithm. In addition, as long as the base algorithm is decentralized, Algorithm 1 will also be decentralized.

## 5 Unknown Non-Stationarity Budget

In this section, we generalize Algorithm 1 to a parameter-free version, which achieves a similar regret bound without the knowledge of the non-stationarity budget and the time horizon $T$. If the

non-stationarity budget is unknown, we cannot determine the appropriate rate to restart in advance as in Algorithm 1. Hence, we use multi-scale testing to monitor the performance of the committed policy and restart adaptively.

## 5.1 Black-box Algorithms for Testing Equilibria

In this section, we present the construction of the testing algorithms TEST_EQ that satisfies Assumption 2 by a black-box reduction to single-agent algorithms, which is able to test whether a policy is an equilibrium in a (near-)stationary game. We make the following assumption on the single-agent learning oracle.

**Assumption 3.** *(PAC guarantee for single-agent RL) We assume that we have access to an oracle* LEARN_OP *such that with probability* $1 - \delta$*, in a single-agent environment with non-stationarity* $\Delta$*, it can output an* $(\epsilon + c_3^\Delta \Delta)$*-optimal policy with* $C_3(\epsilon, \delta)$ *samples.*

The construction of TEST_EQ is described in Protocol 1. We first illustrate how Protocol 1 test NE/CCE in a stationary environment. Note that here we only consider Markov policies and the best response to a Markov policy is the optimal policy in the induced single-agent MDP. First, we sample $\widetilde{O}(\epsilon^{-2})$ trajectories following $\pi$ to get an estimate of $V_i(\pi)$ for all $i$ up to an error bound of $\epsilon/6$ by standard concentration inequalities. Then, for each player $i$, we run LEARN_OP and by Assumption 3, $\pi'_i$ is an $\epsilon/6$-optimal policy in the MDP induced by other players following $\pi_{-i}$. In other words, $\pi'_i$ is an $\epsilon/6$-best response to $\pi_{-i}$. After that we run $(\pi'_i, \pi_{-i})$ for $\widetilde{O}(\epsilon^{-2})$ episodes and estimate the policy value $\widehat{V}_i(\pi'_i, \pi_{-i})$ for players $i$ up to $\epsilon/6$ error bound. Finally the algorithm decides the output according to the empirical estimate of the gap. If the policy is not a $2\epsilon$-EQ, with high probability the empirical gap is larger than $3\epsilon/2$, which leads to a False output. Meanwhile, if the policy is an $\epsilon$-EQ , with high probability the empirical gap is smaller than $3\epsilon/2$, which leads to a True output.

To test a CE, we need to learn the best strategy modification in the induced MDP. While there are many algorithms in prior works that can serve as LEARN_OP, no algorithm is designed for learning the best strategy modification as far as we know. Interestingly, by constructing an MDP with an extended state space, we can reduce learning the best strategy modification to learning the optimal policy in the new MDP. Specifically, here we design an MDP $M'$ such that learning the best strategy modification with random recommendation policy $\pi$ in MDP $M = (\mathcal{S}, \mathcal{A}, P, r, H)$ is equivalent to learning the optimal policy in $M'$, where the randomness in $\pi$ could be correlated with the transition. In $M'$, the state space is $\mathcal{S}' = \mathcal{S} \times \mathcal{A}$, the action space is $\mathcal{A}$, the transition is $P'_h((s_{h+1}, b_{h+1}) \mid (s_h, b_h), a_h) = \mathbb{P}_h(s_{h+1} \mid s_h, \pi_h(s_h) = b_h, a_h) \cdot \pi_{h+1}(b_{h+1} \mid s_{h+1})$ and the reward is $r'_h(\cdot \mid (s_h, b_h), a_h) = r_h(\cdot \mid s_h, \pi_h(s_h) = b_h, a_h)$. The following proposition shows that learning the best strategy modification to recommendation policy $\pi$ in MDP $M$ is equivalent to learning the optimal policy in MDP $M'$. Note that the correlation between $\pi$ and $M'$ complicates the proof of correctness to the seemingly straightforward construction.

**Proposition 2.** *MDP* $M'$ *is induced by MDP* $M$ *and recommendation policy* $\pi$*, then the optimal policy in* $M'$ *corresponds to a best strategy modification to recommendation policy* $\pi$ *in* $M$*.*

Note that the state space in $M'$ is enlarged by a factor of $A$, which means the sample complexity for testing CE is $A$ times larger than CCE, which coincides with the fact that the minimax swap regret is $\sqrt{A}$ times larger than the minimax external regret (Ito, 2020).

**Proposition 3.** *As long as* LEARN_OP *satisfies Assumption 3, Protocol 1 satisfies Assumption 2. Furthermore, there exists algorithm satisfying* $c_2^\Delta = O(H)$*.*

## 5.2 Multi-scale Test Scheduling

In this section, we introduce how to schedule TEST_EQ during the committing phase. The scheduling is motivated by MALG in Wei & Luo (2021), with modifications to the multi-agent setting.

We consider a block of timesteps with length $2^n$ for some integer $n$. The block starts with a LEARN_EQ with $\epsilon = 2^{-n/4}$ and is followed by the committing phase when the agents commit to the learned policy. During the committing phase, TEST_EQ starts randomly for different gaps with different probabilities at each step. That is, we intend to test larger changes more quickly by testing for them more frequently (by setting the probability higher) so that the detection is adaptive to the

---

**Protocol 1** TEST_EQ

1: **Input**: Joint Markov policy $\pi$, failure probability $\delta$, tolerance $\epsilon$.
2: Run $\pi$ for $\widetilde{O}(\epsilon^{-2})$ episodes and estimate the policy value $\widehat{V}_i(\pi)$ with confidence level $\epsilon/6$ for all players $i \in [m]$.
3: **for** $i = 1, 2, \ldots, m$ **do**
4:     Let players $[m]/\{i\}$ follow $\pi_{-i}$ and player $i$ run LEARN_OP with $\delta$ and $\epsilon/6$. Receive best reponse policy $\pi_i'$ or best strategy modification $\psi_i \diamond \pi$ for (NE,CCE) or CE.
5:     Run $(\pi_i', \pi_{-i})$ or $\psi_i \diamond \pi$ for $\widetilde{O}(\epsilon^{-2})$ episodes and estimate the best response value $\widehat{V}_i(\pi_i', \pi_{-i})$ or the best strategy modification value $\widehat{V}_i(\psi_i \diamond \pi)$ with confidence level $\epsilon/6$ for players $i$.
6: **if** $\max_{i \in [m]} \left( \widehat{V}_i(\pi_i', \pi_{-i}) - \widehat{V}_i(\pi) \right) \leq 3\epsilon/2$ or $\max_{i \in [m]} \left( \widehat{V}_i(\psi_i \diamond \pi) - \widehat{V}_i(\pi) \right) \leq 3\epsilon/2$ **then**
7:     **return** True
8: **else**
9:     **return** False

---

severity of changes. Denote the episode index in this block by $\tau$. In the committing phase, if $\tau$ is an integer multiple of $2^{c+q}$ for some $q \in \{0, 1, \cdots, Q\}$, with probability $p(q) = 1/(\epsilon(q)2^{n/2})$ we start a test for gap $\epsilon(q) = \sqrt{c_2/2^q}$ so that the length of test is $2^q$, where $Q, c$ are defined as

$$Q = \min \left\{ \left\lfloor \log_2 \left( c_2 2^{n/2-1} \right) \right\rfloor, n - c \right\}_+, c = \left\lceil 1 + \log_2 \max \left\{ 5\sqrt{c_2}, 2 \log \frac{1}{\delta} \right\} \right\rceil.$$

The gaps we intend to test are approximately $\left\{ \sqrt{2}\epsilon, 2\epsilon, 2\sqrt{2}\epsilon, \cdots \right\}$. It is possible that TEST_EQ for different $\epsilon(q)$ are overlapped. In this case, we prioritize the running of TEST_EQ for larger $\epsilon(q)$ and pause those for smaller $\epsilon(q)$. After the shorter TEST_EQ ends, we resume the longer ones until they are completed. In addition, if a TEST_EQ for $\epsilon(q)$ spans for more than $2^{c+q}$ episodes, it is aborted. To better illustrate the scheduling, we construct an example shown in Figure 2. It can be proved that with high probability no TEST_EQ is aborted (Lemma 5), i.e. the $2^c$ multiplication in length reserves enough space for all TEST_EQ. Note that the original MALG (Wei & Luo (2021)) does not work here because the length of each scheduled TEST_EQ can be reduced greatly and there is no guarantee how a TEST_EQ with reduced length would work. The scheduling is formally stated in Protocol 2.

**Lemma 1.** *With probability* $1 - 3QT\delta$, *the regret inside this block*

$$\text{Regret} = \tilde{O} \left( 2^{3n/4} + c_2 \min \left\{ 2^{2n/3} \left( c_2^{\Delta} \Delta_{[1, E_n]} \right)^{1/3}, 2^{5n/8} \left( c_2^{\Delta} \Delta_{[1, E_n]} \right)^{1/2} \right\} + 2^{n/2} c_2^{3/2} + 2^{-\alpha n/4} c_1 \right). \quad (1)$$

*Remark* 2. The common way to bound the regret with total variation is to divide the block into several near-stationary intervals $[C_1(\epsilon) + 1, 2^n] = \mathcal{I}_1 \cup \mathcal{I}_2 \cup \cdots \cup \mathcal{I}_K$. In each interval the near-stationarity ensure all TEST_EQ to work properly and hence the regret is bounded. This is because if the regret is to big for a long time TEST_EQ would detect it. After that we bound $K$ and finally bound the regret of a block using Hölder's inequality. While prior works (Chen et al., 2019) partition the intervals according to $\Delta_{\mathcal{I}_k} = O\left( |\mathcal{I}_k|^{-1/2} \right)$, we set $\Delta_{\mathcal{I}_k} = O\left( \max \left\{ |I_k|^{-1/2}, 2^{-n/4} \right\} \right)$. This greatly change the subsequent calculations and makes the regret better in our case, please refer to the appendix for more details.

## 5.3 MAIN ALGORITHM

The main algorithm consists of blocks with doubling lengths. The first block is the shortest block that can accomodate a whole LEARN_EQ in it. The doubling structure is not only important to making the algorithm parameter free of $\Delta$, but also to that of $T$ (see Appendix for more details). The performance guarantee of this algorithm is stated in Theorem 1 and Theorem 2. For simplicity, let $\widetilde{\Delta}_{\mathcal{J}} = c_2^{\Delta} \Delta_{\mathcal{J}}$ and $\check{\Delta}_{\mathcal{J}} = \max \left\{ c_1^{\Delta}, c_2^{\Delta} \right\} \Delta_{\mathcal{J}}$.

**Theorem 1.** *With probability* $1 - 3QT\delta$, *the regret of Algorithm 2 is*

$$\text{Regret}(T) = \begin{cases} \tilde{O} \left( \check{\Delta}^{1/5} T^{4/5} + c_2 \min \left\{ \widetilde{\Delta}^{1/3} T^{2/3}, \widetilde{\Delta}^{1/2} T^{5/8} \right\} + \left( c_1 + c_2^{3/2} \right) \check{\Delta}^{2/5} T^{3/5} \right) & \alpha = -2 \\ \tilde{O} \left( c_1 \check{\Delta}^{1/5} T^{4/5} + c_2 \min \left\{ \widetilde{\Delta}^{1/3} T^{2/3}, \widetilde{\Delta}^{1/2} T^{5/8} \right\} + c_2^{3/2} \check{\Delta}^{2/5} T^{3/5} \right) & \alpha = -3 \end{cases}$$

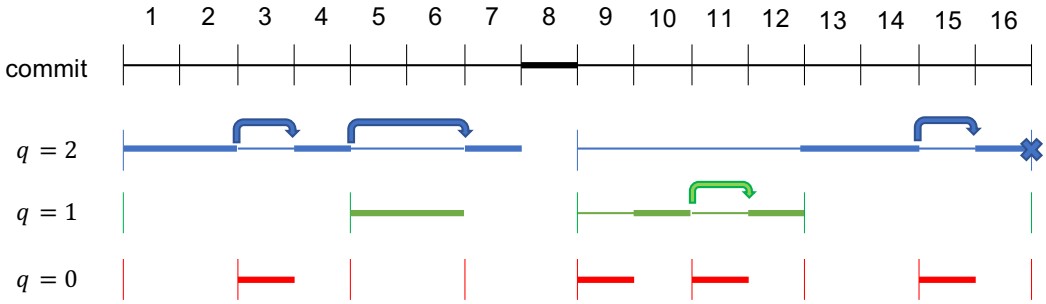

Figure 2: This is an example of the scheduling for committing phase with length $16, Q = 2, c = 1$. The horizontal lines represents the scheduled TEST_EQ except for the black line on the top which represents the time horizon. Different colors represent TEST_EQ for different $\epsilon(q)$. The bold parts of a line represent the active parts and the other parts are the paused parts. The colored vertical lines represent the possible starting points of TEST_EQ for each level. The cross at the last episode indicates the TEST_EQ is aborted because it spans $2^{c+q} = 8$ episodes but has only run $3 < 2^q$ episodes. The bold part of the black line indicates that at this episode we commit to the learned policy. The curved arrow indicates that TEST_EQ is paused and resume later.

---

**Algorithm 2** Multi-scale Testing for Non-stationary MARL

---

1: **Input:** failure probability $\delta$.
2: $N \leftarrow \min \left\{ n \mid 2^n \geq C_1(2^{n/2}) \right\}$
3: **for** $n = N, N + 1, \cdots$ **do**
4:      Schedule a block sized $2^n$ according to Section 5.2 (Protocol 2).
5:      Run LEARN_EQ with accuracy $\epsilon = 2^{-n/4}$ and receive $\pi$.
6:      Run the committing phase according to the schedule. If any TEST_EQ returns False, go to Line 3 and restart the for loop.

---

*Remark* 3. The main idea of the proof is as follows. The restarts divide the whole time horizon into consecutive segments $[1, T] = \mathcal{J}_1 \cup \mathcal{J}_2 \cup \cdots \cup \mathcal{J}_J$. In each segment $\mathcal{J}_j$ between restarts, the regret can be bounded by adding up Formula 1 for all blocks as

$$\text{Regret}\,(\mathcal{J}_j) = \tilde{O}\left( |\mathcal{J}_j|^{3/4} + c_2 \min\left\{ |\mathcal{J}_j|^{2/3} \, \widetilde{\Delta}_{\mathcal{J}_j}^{\,1/3}, |\mathcal{J}_j|^{5/8} \, \widetilde{\Delta}_{\mathcal{J}_j}^{\,1/2} \right\} + c_2^{3/2} \, |\mathcal{J}_j|^{1/2} + c_1 \, |\mathcal{J}_j|^{-\alpha/4} \right).$$

It can be proved that the number of segments is bounded by $J = O\left( T^{1/5} \check{\Delta}^{4/5} \right)$. Using Hölder's inequality, we get the conclusion.

**Theorem 2.** *With probability* $1 - 3QT\delta$, *the regret of Algorithm 2 is*

$$\text{Regret}(T) = \begin{cases} \tilde{O}\left( L^{1/4}T^{3/4} + \left( c_1 + c_2^{3/2} \right) L^{1/2}T^{1/2} \right) & \alpha = -2 \\ \tilde{O}\left( c_1 L^{1/4}T^{3/4} + c_2^{3/2} L^{1/2}T^{1/2} \right) & \alpha = -3 \end{cases}$$

*Remark* 4. Algorithm 2 breaks the curse of multi-agent as long as the base algorithms do. If the base algorithm is decentralized, all players are informed to restart when a change is detected and no further communication is needed. In this sense very few extra communications are needed in Algorithm 2.

## 6 CONCLUSIONS

In this work, we propose black-box reduction approaches for learning equilibria in non-stationary multi-agent reinforcement learning, both with and without knowledge of parameters, that offer favorable performance guarantees. We conclude this paper by posing two open questions. First, it remains unknown how to design algorithms with no-regret oracles, which would further minimize the regret in learning. Second, the lower bound of regret for learning in non-stationary multi-agent systems is currently unknown.

ACKNOWLEDGMENTS

Simon S. Du is supported by NSF IIS 2110170, NSF DMS 2134106, NSF CCF 2212261, NSF IIS 2143493, NSF CCF 2019844, NSF IIS 2229881. Maryam Fazel is supported in part by NSF TRIPODS II-DMS 2023166, NSF CCF 2212261, NSF CCF 2007036, NSF AF 2312775.

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

# A    INTRODUCTION

Multi-agent reinforcement learning (MARL) studies the interactions of multiple agents in an unknown environment with the aim of maximizing their long-term returns (Zhang et al., 2021). This field has applications in diverse areas such as computer games (Vinyals et al., 2019), robotics (de Witt et al., 2020), and smart manufacturing (Kim et al., 2020). Although various algorithms have been developed for MARL, it is typically assumed that the underlying repeated game is stationary throughout the entire learning process. However, this assumption often fails to represent real-world scenarios where the environment is evolving throughout the learning process.

The task of learning within a non-stationary multi-agent system, while crucial, poses additional challenges when attempts are made to generalize non-stationary single-agent reinforcement learning (RL), especially for the bandit feedback case where minimal information is revealed to the agents (Anagnostides et al., 2023). In addition, the various multi-agent settings, such as zero-sum, potential, and general-sum games, along with normal-form and extensive-form games, and fully observable or partially observable Markov games, further complicate the design of specialized algorithms.

In this work, we take the first step towards understanding non-stationary MARL with bandit feedback. First, we point out several challenges that differentiate non-stationary MARL from non-stationary single-agent RL, and bandit feedback from full-information feedback. Subsequently, we propose black-box algorithms with sub-linear dynamic regret in arbitrary non-stationary games, provided there is access to learning algorithms in the corresponding (near-)stationary environment. This versatile approach allows us to leverage existing algorithms for various stationary games, while facilitating seamless adaptation to future algorithms that may offer improved guarantees.

## A.1    RELATED WORK

**(Stationary) Multi-agent reinforcement learning.** Numerous works have been devoted to learning equilibria in (stationary) multi-agent systems, including zero-sum Markov games (Bai et al., 2020; Liu et al., 2021), general-sum Markov games (Jin et al., 2021; Mao et al., 2022; Song et al., 2021; Daskalakis et al., 2022; Wang et al., 2023; Cui et al., 2023), Markov potential games (Leonardos et al., 2021; Song et al., 2021; Ding et al., 2022; Cui et al., 2023), congestion games (Cui et al., 2022), extensive-form games (Kozuno et al., 2021; Bai et al., 2022; Song et al., 2022), and partially observable Markov games (Liu et al., 2022). These works aim to learn equilibria with bandit feedback efficiently, measured by either regret or sample complexity. There also exists a rich literature on asymptotic convergence of different learning dynamics in known games and non-asymptotic convergence with full-information feedback (Swenson et al. (2018); Heliou et al. (2017); Cheung & Piliouras (2020)).

**Non-stationary (single-agent) reinforcement learning.** The study of non-stationary reinforcement learning originated from non-stationary bandits (Auer et al., 2002; Besbes et al., 2014; Chen et al., 2019; Zhao et al., 2020; Wei & Luo, 2021; Cheung et al., 2022; Garivier & Moulines, 2011). Auer et al. (2019) and Chen et al. (2019) first achieve near-optimal dynamic regret without knowing the non-stationary budget. For non-stationary RL, Cheung et al. (2020) develop a Bandit-over-Reinforcement-Learning framework for non-stationary RL that generalizes the Bandit-over-Bandit approach (Cheung et al., 2022). The most relevant work is Wei & Luo (2021), which also proposes a black-box approach with multi-scale testing and achieves optimal regret in various single-agent settings. We refer readers to Wei & Luo (2021) for a more comprehensive literature review on non-stationary reinforcement learning.

**Non-stationary multi-agent reinforcement learning.** Most of the previous works have been focused on the full-information feedback setting, which is considerably easier than the bandit feedback setting as testing becomes unnecessary (Cardoso et al., 2019; Zhang et al., 2022; Anagnostides et al., 2023; Duvocelle et al., 2022; Poveda et al., 2022). For two-player zero-sum matrix games, Zhang et al. (2022) proposes a meta-algorithm over a group of base algorithms to tackle with unknown parameters. Anagnostides et al. (2023) studies the convergence of no-regret learning dynamics in non-stationary matrix games, including zero-sum, general-sum and potential games, and shares a similar dynamic regret notion as ours. Notably, Cardoso et al. (2019) also studies the bandit feedback case and aims to minimize NE-regret, while the regret is comparing with the best NE in hindsight instead of a dynamic regret.

## B    CHALLENGES IN NON-STATIONARY GAMES

In this section, we discuss the challenges in non-stationary games in more detail.

### B.1    CHALLENGES IN TEST-BASED ALGORITHMS

The idea of achieving optimal regret using consecutive testing in a parameter-free fashion was first proposed in Auer et al. (2019). Here we restate the idea as follows. Consider the multi-armed bandit setting. There are $K$ arms, $T$ episodes and $L$ abrupt changes. The regret can be decomposed as

- Most of the time, we run the standard UCB algorithm. If we always restart the UCB algorithm right after each abrupt change, the accumulated regret is upper bounded by $O\left(\sqrt{K(T/L)}L\right) = O\left(\sqrt{KTL}\right)$.

- Intending to detect changes on one arm that make the optimal arm incur $D$ regret, the algorithm starts a test at each step with probability $p_D = D\sqrt{l/KT}$ where $l$ is the number of changes detected thus far. The test should last $n_D = O(1/D^2)$ steps to make the confidence bound no larger than $D$. In expectation, the test incurs $p_D T n_D \Delta = O\left(\frac{\Delta}{D}\sqrt{\frac{lT}{K}}\right)$ regret.

  Here $\Delta$ is the real gap of the detected arm. To cover all possible $\Delta$, we may detect for gaps of size $D = D_0, 2D_0, 4D_0, \cdots$. $D_0$ is the smallest gap that is worth noticing[4]. This incurs $O\left(\sqrt{\frac{LT}{K}}K\right) = O\left(\sqrt{KTL}\right)$ regret.

- The expected number of episodes before we start to detect for a change of size $D$ is $D/p_D = \sqrt{KT/l}$. Summing over all changes, this part incurs $O\left(KTL\right)$ regret

In all, the scheme suffer $O\left(\sqrt{KLT}\right)$ regret, which is optimal. In the game setting, however, the second part can become $\frac{1}{D_0}\sqrt{\frac{lT}{K}}K$ and we will no longer have a no-regret algorithm.

### B.2    CHALLENGES IN BANDIT-OVER-RL ALGORITHMS

The high-level idea of BORL is as follows (Cheung et al., 2020). First partition the whole time horizon $T$ into intervals with length $H$. Each interval is one step for an adversarial bandit algorithm $A$. Inside each interval, one instance of the base algorithm is run, with the tunable parameter selected by $A$. The arms for $A$ are the possible parameters of the base algorithm and the reward is the total reward from one interval. Let the action at timestep $t$ be $a_t$ and $r(a_t)$ be its expected reward, $a_t^*$ be the optimal action at timestep $t$ and $R(w)$ be the expected return from one interval if we chooses parameter $w$. The regret can then be decomposed as

$$\sum_{t=1}^{T}\left[r\left(a_t^*\right) - r\left(a_t\right)\right] = \left[\sum_{t=1}^{T}r\left(a_t^*\right) - \sum_{h=1}^{T/H}R(w_h)\right] + \left[\sum_{h=1}^{T/H}R(w_h) - \sum_{t=1}^{T}r(a_t)\right]$$

where $w_h$ is the best parameter in interval $h$. The first term is bounded by the base algorithm regret upper bound and the second term is bounded by the adversarial bandit regret guarantee. If we apply the same to minimize, for example, the Nash regret

$$\sum_{t=1}^{T}\max_{i\in[m]}\left(V_i^M(\dagger, \pi_{-i}) - V_i^M(\pi)\right),$$

we easily find the max hinders the same decomposition. Even if we drop the max and focus on individual regret, the decomposition is

$$\sum_{t=1}^{T}\left[V_i^M(\dagger, \pi_{-i}) - V_i^M(\pi)\right] = \left[\sum_{i=1}^{T}V_i^M(\dagger, \pi_{-i}) - \sum_{h=1}^{T/H}R(w_h)\right] + \left[\sum_{h=1}^{T/H}R(w_h) - \sum_{t=1}^{T}V_i^M(\pi)\right]$$

---

[4]We can take $D_0 = \sqrt{K/T}$ because even if each step we suffer an extra $D_0$ regret, the total regret will still remain.

where the first term loses meaning. The fundamental reason is that in MAB, at timestep $t$, any action is competing with a fixed action $a_t^*$, while in a game, a policy $\pi$ is competing with $\arg\max_{\pi_i'} V_i^M(\pi_i', \pi_{-i})$, which depends on $\pi$ itself. This difficulty can also be seen from Figure 1.

## C  OMITTED PROOFS IN SECTION 4

In this section, we analyze the performance of Algorithm 1. For convenience, we denote the intervals corresponding to each LEARN_EQ by $\mathcal{I}_1, \mathcal{I}_2, \cdots, \mathcal{I}_K$ and the committing phases as $\mathcal{J}_1, \mathcal{J}_2, \cdots, \mathcal{J}_K$. The committed policy are $\pi^1, \pi^2, \cdots, \pi^K$ respectively. Here $K = \lceil T/(C_1(\epsilon) + T_1) \rceil$ and $\mathcal{J}_K$ can be empty.

**Lemma 2.** *If $x > 1$, $x/2 < \lceil x \rceil < x + 1$.*

*Remark* 5. It is a basic algebraic lemma that will be used very often to get over the roundings.

**Lemma 3.** *If $\pi$ is an $\epsilon$-EQ of episode $t$, then it is also an $\left(\epsilon + 2H\Delta_{[t,t']}\right)$-equilibrium for any episode $t' > t$.*

*Proof.* To facilitate this proof, we define some more notations. The value function of player $i$ at timestep $h_0$, episode $t$, state $s$ is defined to be

$$V_{h_0,i}^{\pi,M}(s) = \mathbb{E}_\pi \left[ \sum_{h=h_0}^{H} r_{h,i}(s_h, \mathbf{a}_h) \mid M, s_{h_0} = s \right]. \tag{2}$$

Here $M$ is the model at episode $t$. We also denote the model at episode $t'$ by $M'$. We have the recursion

$$V_{h_0,i}^{\pi,M}(s) = \sum_{\mathbf{a}} \pi(\mathbf{a} \mid s) \left[ \sum_{s'} \mathbb{P}_{h_0}^M(s'|s,\mathbf{a}) V_{h_0+1,i}^{\pi,M}(s') + R_{h_0,i}^M(s,\mathbf{a}) \right].$$

Assume

$$\left| V_{h_0+1,i}^{\pi,M}(s) - V_{h_0+1,i}^{\pi,M'}(s) \right| \le H \sum_{h=h_0+1}^{H} \left( \left\| \mathbb{P}_h^M - \mathbb{P}_h^{M'} \right\|_1 + \left\| R_h^M - R_h^{M'} \right\|_1 \right),$$

then we have

$$\left| V_{h_0,i}^{\pi,M}(s) - V_{h_0,i}^{\pi,M'}(s) \right|$$

$$\le \sum_{\mathbf{a}} \pi(\mathbf{a} \mid s) \left[ \sum_{s'} \left( \mathbb{P}_{h_0}^M(s'|s,\mathbf{a}) - \mathbb{P}_{h_0}^{M'}(s'|s,\mathbf{a}) \right) V_{h_0+1,i}^{\pi,M'}(s') \right]$$

$$+ \sum_{\mathbf{a}} \pi(\mathbf{a} \mid s) \left[ \sum_{s'} \mathbb{P}_{h_0}^M(s'|s,\mathbf{a}) \left( V_{h_0+1,i}^{\pi,M'}(s') - V_{h_0+1,i}^{\pi,M}(s') \right) \right]$$

$$+ \sum_{\mathbf{a}} \pi(\mathbf{a} \mid s) \left[ R_{h_0,i}^M(s,\mathbf{a}) - R_{h_0,i}^{M'}(s,\mathbf{a}) \right]$$

$$\le H \left| \mathbb{P}_{h_0}^M(s'|s,\mathbf{a}) - \mathbb{P}_{h_0}^{M'}(s'|s,\mathbf{a}) \right| + \left| V_{h_0+1,i}^{\pi,M}(s) - V_{h_0+1,i}^{\pi,M'}(s) \right| + \left| R_{h_0,i}^M(s,\mathbf{a}) - R_{h_0,i}^{M'}(s,\mathbf{a}) \right|$$

$$\le H \sum_{h=h_0}^{H} \left( \left\| \mathbb{P}_h^M - \mathbb{P}_h^{M'} \right\|_1 + \left\| R_h^M - R_h^{M'} \right\|_1 \right).$$

Since the assumption holds trivially for $h_0 = H$, by induction we get

$$\left| V_1^{\pi,M}(s) - V_1^{\pi,M'}(s) \right| \le \Delta_{[t,t']}.$$

Finally by definition of the equilibria, we get the conclusion. $\qquad\square$

**Lemma 4.** *With probability $1 - T\delta$, $\pi^k$ is $\left(\epsilon + c_1^\Delta \Delta_{\mathcal{I}_k}\right)$-approximate equilibrium in the last episode of $\mathcal{I}_k$ for all $k \in [K]$.*

*Proof.* This is by the union bound and $K \leq T$. $\qquad\square$

The following theorem is conditioned on this high-probability event.

**Proposition 1.** *With probability* $1 - T\delta$, *the regret of Algorithm 1 satisfies*

$$\text{Regret}(T) \leq \frac{4TC_1(\epsilon)}{T_1} + T\epsilon + 2\max\left\{c_1^\Delta, H\right\}T_1\Delta.$$

*Proof.* According to Assumption 1, $\pi^k$ is an $\left(\epsilon + c_1^\Delta \Delta_{\mathcal{I}_k}\right)$-approximate equilibrium for the last episode of $\mathcal{I}_k$. Hence it is an $\left(\epsilon + 2\max\left\{c_1^\Delta, H\right\} \Delta_{\mathcal{I}_k \cup \mathcal{J}_k}\right)$-approximate equilibrium for any episode in $\mathcal{J}_k$ according to Lemma 3. In the proof we omit the max with $H$ and recover it in the conclusion.

$$\begin{aligned}
\text{Regret}(T) &= \sum_{k=1}^{K} \left(|\mathcal{I}_k| + |\mathcal{J}_k|\left(\epsilon + 2c_1^\Delta \Delta_{\mathcal{I}_k \cup \mathcal{J}_k}\right)\right)\\
&\leq K\lceil C_1(\epsilon)\rceil + T\epsilon + 2c_1^\Delta T_1 \Delta\\
&\leq \frac{4TC_1(\epsilon)}{T_1} + T\epsilon + 2c_1^\Delta T_1 \Delta.
\end{aligned}$$

$\qquad\square$

**Corollary 1.** *With probability* $1 - T\delta$, *the regret of Algorithm 1 satisfies*

$$\text{Regret}(T) \leq \begin{cases}
13\left(\Delta c_1 \max\left\{c_1^\Delta, H\right\}\right)^{1/4} T^{3/4}, & \alpha = -2,\\
13\left(\Delta c_1 \max\left\{c_1^\Delta, H\right\}\right)^{1/5} T^{4/5}, & \alpha = -3,
\end{cases}$$

*by setting*

$$T_1 = \left\lceil \sqrt{\frac{TC_1(\epsilon)}{\max\left\{c_1^\Delta, H\right\}\Delta}} \right\rceil, \quad \epsilon = \begin{cases}
\left(\Delta c_1 \max\left\{c_1^\Delta, H\right\}/T\right)^{1/4}, & \alpha = -2,\\
\left(\Delta c_1 \max\left\{c_1^\Delta, H\right\}/T\right)^{1/5}, & \alpha = -3.
\end{cases}$$

*Proof.* As before, we omit the max with $H$ in the proof.

$$\begin{aligned}
\text{Regret}(T) &\leq 8TC_1(\epsilon)\sqrt{\frac{c_1^\Delta \Delta}{TC_1(\epsilon)}} + T\epsilon + 4c_1^\Delta \Delta \sqrt{\frac{TC_1(\epsilon)}{c_1^\Delta \Delta}}\\
&= 12\sqrt{c_1^\Delta T\Delta C_1(\epsilon)} + T\epsilon\\
&= 12\sqrt{c_1^\Delta T\Delta c_1}\epsilon^{\alpha/2} + T\epsilon
\end{aligned}$$

Applying Lemma 2, we get the desired conclusion. $\qquad\square$

## D   OMITTED PROOFS IN SECTION 5

In Section D.1 we present the proof for Proposition 2 and Proposition 3. In Section D.2 and D.3, we analyze the performance of Algorithm 2. We first analyze the performance of single block in Section D.2 and then present the subsequent proof in Section D.3. For convenience, the episodes in Section D.2 refer to $\tau$ and the episodes in Section D.3 refer to $t$. Before presenting the proof, we first describe the multi-scale scheduling formally in Protocol 2.

### D.1   PROOFS REGARDING CONSTRUCTION OF TEST_EQ

**Proposition 2.** *MDP* $M'$ *is induced by MDP* $M$ *and recommendation policy* $\pi$, *then the optimal policy in* $M'$ *corresponds to a best strategy modification to recommendation policy* $\pi$ *in* $M$.

---

**Protocol 2** Scheduling TEST_EQ in a block with length $2^n$

---

1: **Input**: Joint Markov policy $\pi$, failure probability $\delta$, tolerance $\epsilon$.
2: **for** $\tau = 0, 1, \ldots, 2^n - 1$ **do**
3:     **for** $q = 0, 1, \cdots, Q$ **do**
4:         **if** $\tau$ is a multiple of $2^{c+q}$ **then**
5:             With probability $p(q)$, schedule a TEST_EQ for $\epsilon(q)$ starting from $\tau$.

---

*Proof.* To facilitate the proof, we define some notations here. We define the value function of policy $\pi$ in an MDP $M$ at timestep $h_0$ and state $s$ as

$$V_{h_0}^{\pi,M}(s) = \mathbb{E}_\pi \left[ \sum_{h=h_0}^{H} r_h(s_h, a_h) \mid M, s_{h_0} = s \right].$$

The mean reward from $r_h(\cdot|s,a)$ is denoted as $R_h(s,a)$. Let $\pi'$ be a policy in $M$, then

$$V_h^{\pi',M'}((s,b)) = \sum_a \pi'(a \mid (s,b)) \left[ \sum_{(s',b')} P_h'\left((s',b')|(s,b),a\right) V_{h+1}^{\pi',M'}\left((s',b')\right) + R_h'((s,b),a) \right]$$

Additionally, the Q-function of a state-action pair $(s,b)$ under policy $\pi$ at timestep $h_0$ for agent $i$ in Markov game $M$ is defined as

$$Q_{h_0,i}^{\pi,M}(s,b) = \mathbb{E}_\pi \left[ \sum_{h=h_0}^{H} r_{h,i}(s_h, \mathbf{a}_h) \mid M, s_{h_0} = s, a_{h_0,i} = b \right].$$

Assume $\pi'$ is a deterministic policy and $\psi_i$ is a strategy modification such that its choice is the same as the choice of $\pi'$, then

$$Q_{h_0,i}^{\psi_i \diamond \pi,M}(s, \psi_i(b)) = \sum_{(s',b')} \mathbb{P}_h(s'|s, \pi_h(s) = b, \psi_i(b)) \pi_{h+1}(b' \mid s') Q_{h_0+1,i}^{\psi_i \diamond \pi,M}(s', \psi_i(b))$$
$$+ R_{h_0}(s, \psi_i(b) \mid \pi_h(s) = b)$$

by definition of $M'$ we can directly see that

$$V_h^{\pi',M'}((s,b)) = Q_{h,i}^{\psi_i \diamond \pi,M}(s, \psi_i(b)) \tag{3}$$

Hence the optimal policy of $M'$ corresponds to a best strategy modification to recommendation policy. $\qquad\square$

**Proposition 3.** *As long as* LEARN_OP *satisfies Assumption 3, Protocol 1 satisfies Assumption 2. Furthermore, there exists algorithm satisfying $c_2^\Delta = O(H)$.*

*Proof.* We first consider the NE and CCE case. The main logic has been stated in the main text. We restate it here with environmental changes involved. Denote the intervals that run Line 2, 4, 5 by $\mathcal{I}, \mathcal{J}, \mathcal{K}$ respectively. Then with high probability, the estimation of $\widehat{V}_i(\pi)$ departs from the true value by at most $\epsilon/6 + \Delta_\mathcal{I}$ and that of $\widehat{V}_i(\pi_i', \pi_{-i})$ is at most $\epsilon/3 + c_3^\Delta \Delta_\mathcal{J} + \Delta_\mathcal{K}$. Combine all the error we get the conclusion. In terms of sample complexity

$$C_2(\epsilon) = \widetilde{O}\left(mC_3(\epsilon) + \epsilon^{-2}\right) = \widetilde{O}\left(mC_3(\epsilon)\right).$$

The last equality use the information-theoretic lower bound $C_3(\epsilon) = \Omega(\epsilon^{-2})$. Then we consider the CE case. By Equation 3 we can prove the correctness of this algorithm using the same argument as before. In terms of sample complexity, it is the same as before except that we need to change the size of state space from $|\mathcal{S}|$ to $|\mathcal{S}||\mathcal{A}|$. Finally, $c_2^\Delta = c_3^\Delta$. By Wei & Luo (2021), we know that we have $c_2^\Delta = c_3^\Delta = O(H^2)$. $\qquad\square$

## D.2 SINGLE BLOCK ANALYSIS

Divide $[C_1(\epsilon) + 1, 2^n]$ into $\mathcal{I}_1 \cup \mathcal{I}_2 \cup \cdots \cup \mathcal{I}_K$ such that $\mathcal{I}_k = [s_k, e_k], s_1 = C_1(\epsilon) + 1, e_K = 2^n, e_k + 1 = s_{k+1}$ and

$$\Delta_{\mathcal{I}_k} \leq \frac{1}{c_2^\Delta} \max\left\{\frac{1}{\sqrt{|\mathcal{I}_k|}}, 2^{-n/4-1}\right\}$$

Intervals with such property are called near-stationary. Let $E_n \in \mathcal{I}_l$ be the last episode (The block may be ended due to a failed TEST_EQ). Define $e'_k = \min\{E_n, e_k\}, \mathcal{I}'_k = [s_i, e'_k]$. If $k > l$, $\mathcal{I}'_k = \varnothing$. For convenience, we denote $\tau_n = C_1(\epsilon) + 1$ in the following proof.

**Definition 6.** *For $k \in [K], q \in \{0, 1, \cdots, Q\}$, let*

$$\tau_k(q) = \min\left\{\tau \in \mathcal{I}'_k \mid \pi \text{ is not a } 2\epsilon(q)\text{-EQ at } \tau\right\}, \xi_k(q) = \left[e'_k - \tau_k(q) + 1\right]_+.$$

First, we are going to show that with high probability no TEST_EQ is aborted.

**Lemma 5.** *With probability $1 - 2QT\delta$, for any TEST_EQ instance testing gap $\epsilon(q)$ maintained from $s$ to $e$, it returns fail if the policy is not $(2\epsilon(q) + c_2^\Delta \Delta_{[s,e]})$-NE/CCE for any $\tau \in [s, e]$. In equivalence, $e - s < 2^{c+q}$ and all TEST_EQ function as desired.*

*Proof.* By union bound, the probability all TEST_EQ function as desired is $1 - QT\delta$. There are $2^{q-r}$ possible starting points for a test occupying $2^r$ episodes. For each of them, TEST_EQ exists with probability $1/(\epsilon(r)2^{n/2})$. By Bernstein's inequality, with probability $1 - \delta$, the number of such tests is upper-bounded by

$$2^{q-r}\frac{1}{\epsilon(r)2^{n/2}} + \sqrt{2 \cdot 2^{q-r}\frac{1}{\epsilon(r)2^{n/2}} \log\frac{1}{\delta}} + \log\frac{1}{\delta}$$

$$\leq 2 \cdot 2^{q-r}\frac{1}{\epsilon(r)2^{n/2}} + 2\log\frac{1}{\delta}$$

$$= \frac{2^{q-r/2+1}}{\sqrt{c_2}2^{n/2}} + 2\log\frac{1}{\delta}.$$

By union bound, this inequality holds for all TEST_EQ with probability $1 - QT\delta$. So the total length of all shorter tests is upper bounded by

$$\sum_{r=0}^{q-1}\left(\frac{2^{q-r/2+1}}{\sqrt{c_2}2^{n/2}} + 2\log\frac{1}{\delta}\right)2^r$$

$$\leq 2^{q+1}\frac{2^{\frac{q-1}{2}} - 1}{\sqrt{2} - 1}\frac{1}{\sqrt{c_2}2^{n/2}} + 2\log\frac{1}{\delta}(2^q - 1)$$

$$\leq 5\sqrt{c_2}\left(2^{\frac{q-1}{2}} - 1\right) + \log\frac{1}{\delta}2^{q+1}$$

$$\leq \max\left\{5\sqrt{c_2}, 2\log\frac{1}{\delta}\right\}2^q$$

Here we use $2^q < 2^Q < c_2 2^{n/2}$. Using the union bound, we get the conclusion. $\qquad\square$

In subsequent proofs, we condition on the high probability event described in this lemma.

**Lemma 6.** *With probability $1 - Q\delta$, for all $r \in [Q]$.*

$$\sum_{k=1}^{l}\left[\xi_k(r) - 2^{c+r}\right]_+ \leq 2^{c+r-1}\epsilon(r-2)\sqrt{2^n}\log\frac{1}{\delta} = 2^c\sqrt{2^{r+n}c_2}$$

*Proof.* For each $r \in [Q]$.

$$2^{-c-r+2}\sum_{k=1}^{l}\left[\xi_k(r) - 2^{c+r}\right]_+$$

$$= 2^{-c-r+2} \sum_{k=1}^{l} \left[ e'_k - \tau_k(r) + 1 - 2^{c+r} \right]_+$$

$$\leq \sum_{k=1}^{K} \sum_{\tau \in \mathcal{I}_k} \mathbb{1} \left[ \tau \in [\tau_k(r), e'_k - 2^{c+r-1}], \tau \bmod 2^{c+r-2} \equiv 0 \right]$$

$$= \sum_{\tau = \tau_n}^{2^n} \mathbb{1} \left[ \tau \in [\tau_k(r), e'_k - 2^{c+r-1}], \tau \bmod 2^{c+r-2} \equiv 0 \right]$$

$$\leq \sum_{\tau = \tau_n}^{2^n} \mathbb{1} \left[ \tau \in [\tau_k(r), e'_k - 2^{c+r-1}], \tau \bmod 2^{c+r-2} \equiv 0 \text{ and there is no test for } \epsilon(r)/2 \text{ starting at any } t \in [\tau_n, \tau] \right]$$

$$+ \sum_{\tau = \tau_n}^{2^n} \mathbb{1} \left[ \tau \in [1, E_n - 2^{c+r-1}] \text{ and there is a test for } \epsilon(r)/2 \text{ starting at some } t \in [\tau_n, \tau] \right]$$

$$\leq \left[ 1 + \frac{\log(1/\delta)}{-\log(1 - 1/(\epsilon(r-2)\sqrt{2^n}))} \right] + 0 \leq 2\epsilon(r-2)\sqrt{2^n} \log \frac{1}{\delta}$$

The first inequality holds because in an interval of length $w$, there are at least $(w + 2 - 2u)/u$ points whose indices are multiples of $u$. The third inequality holds with probability $1 - \delta$. The first sum is bounded using the fact the test is started i.i.d. with constant probability $1/(\epsilon(r-2)\sqrt{2^n})$. In the second sum, the condition implies that the ending time of the test is before $t + 2^{c+r-2} - 1 \leq e_i - 2^{c+r-2} - 1 \leq e_i$ so the test is within $\mathcal{I}_k$ and $t + 2^{c+r-2} - 1 \leq \tau + 2^{c+r-2} - 1 < E_n$ so the test ends before the block ends. However, the test is for $\epsilon(r)$ and the variation during the test is bounded by $\Delta_{\mathcal{I}_k} < 2^{-n/4} = \epsilon < \epsilon(r)$, so such TEST_EQ must return Fail. □

In subsequent proofs, we further condition on the high probability event described in this lemma.

**Lemma 7.** *The total number of near-stationary intervals*

$$l \leq 1 + 2 \min \left\{ 2^{n/3} \left( c_2^\Delta \Delta_{[1,E_n]} \right)^{2/3}, 2^{n/4} c_2^\Delta \Delta_{[1,E_n]} \right\} \tag{4}$$

*Proof.* We divide $[\tau_n, E_n] = \mathcal{I}_1 \cup \mathcal{I}_2 \cup \cdots \cup \mathcal{I}_l$ in such a way that $[s_k, e_k]$ is near-stationary but $[s_k, e_k + 1]$ is not near-stationary. Then

$$\Delta_{[\tau_n, E_n]} \geq \sum_{k=1}^{l-1} \Delta_{[s_k, e_k+1]}$$

$$\geq \frac{1}{c_2^\Delta} \sum_{k=1}^{l-1} \max \left\{ \frac{1}{\sqrt{e_k - s_k + 2}}, 2^{-n/4-1} \right\}$$

$$\geq \frac{1}{c_2^\Delta} \max \left\{ \sum_{k=1}^{l-1} \frac{1}{2\sqrt{e_k - s_k + 1}}, (l-1)2^{-n/4-1} \right\}$$

Hence by Hölder's inequality

$$l \leq 1 + \min \left\{ \left( \sum_{k=1}^{l-1} (e_k - s_k + 1)^{-1/2} \right)^{2/3} \left( \sum_{k=1}^{l-1} (e_k - s_k + 1) \right)^{1/3}, 2^{n/4+1} c_2^\Delta \Delta_{[\tau_n, E_n]} \right\}$$

$$\leq 1 + 2 \min \left\{ \left( c_2^\Delta \Delta_{[\tau_n, E_n]} \right)^{2/3} |[\tau_n, E_n]|^{1/3}, 2^{n/4} c_2^\Delta \Delta_{[\tau_n, E_n]} \right\}$$

$$\leq 1 + 2 \min \left\{ 2^{n/3} \left( c_2^\Delta \Delta_{[1,E_n]} \right)^{2/3}, 2^{n/4} c_2^\Delta \Delta_{[1,E_n]} \right\}$$

□

**Lemma 8.** *With probability* $1 - 3QT\delta$

$$\text{Regret}([1, E_n]) \leq 2^{3n/4+4} + 4Q \left( 2^{n/2+c}\sqrt{c_2 l} + 2^{c+n/2}c_2 \right) + c_2 \log \frac{1}{\delta} 2^{n/2+1} + c_1 2^{-\alpha n/4}$$

*Proof.* First we consider the regret generated by TEST_EQ. We need to count the number of steps all the tests go for. Similar to the calculation in Lemma 6. The number of tests with length $2^q$ is upper bounded by

$$\frac{2^{n-r/2+1}}{c\sqrt{c_2}2^{n/2}} + 2\log\frac{1}{\delta}.$$

So the total length of all TEST_EQ is upper bounded by

$$\sum_{r=0}^{Q}\left(\frac{2^{n-r/2+1}}{2^c\sqrt{c_2}2^{n/2}} + 2\log\frac{1}{\delta}\right)2^r$$

$$\leq 2^{n+1}\frac{2^{Q/2}-1}{\sqrt{2}-1}\frac{1}{c\sqrt{c_2}2^{n/2}} + 2\log\frac{1}{\delta}\left(2^Q-1\right)$$

$$\leq \frac{5}{2^c}2^{3n/4} + c_2\log\frac{1}{\delta}2^{n/2+1}$$

Then we consider the regret generated by committing.

$$\sum_{\tau\in\mathcal{I}'_k}\mathrm{Gap}^{M^t}\left(\pi^t\right)$$

$$\leq \sum_{\tau\in\mathcal{I}'_k}\left(\mathbb{1}\left[\mathrm{Gap}^{M^t}\left(\pi^t\right)\leq 2\epsilon(Q)\right]2\epsilon(Q)\right.$$

$$\left.+ \sum_{r=0}^{Q-1}\mathbb{1}\left[2\epsilon(r+1)\leq\mathrm{Gap}^{M^t}\left(\pi^t\right)\leq 2\epsilon(r)\right]2\epsilon(r) + \mathbb{1}\left[\mathrm{Gap}^{M^t}\left(\pi^t\right)>\epsilon(0)\right]1\right)$$

$$\leq 2|\mathcal{I}'_k|\epsilon(Q) + 2\sum_{r=0}^{Q-1}\epsilon(r)\xi_i(r+1) + 2\epsilon(0)\xi_i(0)$$

$$\leq 2|\mathcal{I}'_k|\epsilon(Q) + 4\sum_{r=0}^{Q}\epsilon(r)\xi_i(r)$$

In the second inequaility we use $\epsilon(0) = \sqrt{c_2} > 1$ and in the third inequality we use $\epsilon(r) \leq 2\epsilon(r+1)$. Summing over all intervals we have

$$\mathrm{Regret}([1,E_n]) \leq 2^{n+1}\epsilon(Q) + 4\sum_{r=0}^{Q}\sum_{k=1}^{l}\epsilon(r)\xi_k(r).$$

Furthermore

$$\sum_{k=1}^{l}\epsilon(r)\xi_k(r) = \sum_{k=1}^{l}\epsilon(r)\min\{\xi_k(r),2^{c+r}\} + \sum_{k=1}^{l}\epsilon(r)\left[\xi_k(r)-2^{c+r}\right]_+$$

$$\leq 2^c\sum_{k=1}^{l}\sqrt{c_2\min\{\xi_k(r),2^{c+r}\}} + \sum_{k=1}^{l}\epsilon(r)\left[\xi_k(r)-2^{c+r}\right]_+$$

$$\leq 2^c\sum_{k=1}^{l}\sqrt{c_2\left|\mathcal{I}'_k\right|} + 2^{c+n/2}c_2$$

The last inequality uses Lemma 6. Hence

$$\mathrm{Regret}([1,E_n]) \leq 2^{n+1}\epsilon(Q) + 4Q\left(2^c\sum_{k=1}^{l}\sqrt{c_2\left|\mathcal{I}'_k\right|} + 2^{c+n/2}c_2\right) + \frac{5}{2^c}2^{3n/4} + c_2\log\frac{1}{\delta}2^{n/2+1} + C_1(\epsilon)$$

$$\leq 2^{3n/4+4} + 4Q\left(2^c\sqrt{c_2l\sum_{k=1}^{l}|\mathcal{I}'_k|} + 2^{c+n/2}c_2\right) + c_2\log\frac{1}{\delta}2^{n/2+1} + c_12^{-\alpha n/4}$$

$$\leq 2^{3n/4+4} + 4Q\left(2^{n/2+c}\sqrt{c_2l} + 2^{c+n/2}c_2\right) + c_2\log\frac{1}{\delta}2^{n/2+1} + c_12^{-\alpha n/4}$$

$$\square$$

To keep the notation clean, from now on we make frequent use of the big-O notation and hide the dependencies on logarithmic factors on relevant variables. We also assume $\Delta$ is always large enough so that we can drop the 1 in Inequality 4.

**Lemma 1.** *With probability $1 - 3QT\delta$, the regret inside this block*

$$\text{Regret} = \tilde{O}\left( 2^{3n/4} + c_2 \min\left\{ 2^{2n/3}\left(c_2^\Delta \Delta_{[1,E_n]}\right)^{1/3}, 2^{5n/8}\left(c_2^\Delta \Delta_{[1,E_n]}\right)^{1/2} \right\} + 2^{n/2}c_2^{3/2} + 2^{-\alpha n/4}c_1 \right). \quad (1)$$

*Proof.* We may restate the bounds in Lemma 7 and 8 as

$$l = O\left( \min\left\{ 2^{n/3}\left(c_2^\Delta \Delta\right)^{2/3}_{[1,E_n]}, 2^{n/4}\left(c_2^\Delta \Delta_{[1,E_n]}\right) \right\} \right)$$

$$\text{Regret}([1, E_n]) = \tilde{O}\left( 2^{3n/4} + 2^{n/2}c_2\sqrt{l} + 2^{n/2}c_2^{3/2} + 2^{-\alpha n/4}c_1 \right)$$

Combine them together we get

$$\text{Regret}([1, E_n]) = \tilde{O}\left( 2^{3n/4} + c_2 \min\left\{ 2^{2n/3}\left(c_2^\Delta \Delta_{[1,E_n]}\right)^{1/3}, 2^{5n/8}\left(c_2^\Delta \Delta_{[1,E_n]}\right)^{1/2} \right\} + 2^{n/2}c_2^{3/2} + 2^{-\alpha n/4}c_1 \right)$$
$$(5)$$

$\square$

### D.3 PROOF FOR THEOREM 1

Due to the doubling structure inside each segment, from Formula 5 we get

$$\text{Regret}\left(\mathcal{J}_j\right) = \tilde{O}\left( |\mathcal{J}_j|^{3/4} + c_2 \min\left\{ |\mathcal{J}_j|^{2/3}\left(c_2^\Delta \Delta_{\mathcal{J}_j}\right)^{1/3}, |\mathcal{J}_j|^{5/8}\left(c_2^\Delta \Delta_{\mathcal{J}_j}\right)^{1/2} \right\} + c_2^{3/2}|\mathcal{J}_j|^{1/2} + c_1 |\mathcal{J}_j|^{-\alpha/4} \right)$$

**Lemma 9.**

$$J = O\left( T^{1/5}\left( \max\left\{ c_1^\Delta, c_2^\Delta \right\} \Delta \right)^{4/5} \right).$$

*Proof.* For any segment $\mathcal{J}_j$,

$$\max\left\{ c_1^\Delta, c_2^\Delta \right\} \Delta_{\mathcal{J}_j} \geq \epsilon(Q) - \epsilon \geq \left( \sqrt{2} - 1 \right) |\mathcal{J}_j|^{-1/4}$$

since the ending of a segment is caused by a False returned by TEST_EQ. Then by the same logic as in Lemma 7 we get the conclusion $\square$

Hence by Hölder inequality

$$\text{Regret}(T) = \tilde{O}\left( J^{1/4}T^{3/4} + c_2 \min\left\{ T^{2/3}\widetilde{\Delta}^{1/3}, T^{5/8}\widetilde{\Delta}^{1/2} \right\} + c_2^{3/2}J^{1/2}T^{1/2} + c_1 J^{1+\alpha/4}T^{-\alpha/4} \right)$$
$$= \begin{cases} \tilde{O}\left( \check{\Delta}^{1/5}T^{4/5} + c_2 \min\left\{ \widetilde{\Delta}^{1/3}T^{2/3}, \widetilde{\Delta}^{1/2}T^{5/8} \right\} + \left(c_1 + c_2^{3/2}\right)\check{\Delta}^{2/5}T^{3/5} \right) & \alpha = -2 \\ \tilde{O}\left( c_1\check{\Delta}^{1/5}T^{4/5} + c_2 \min\left\{ \widetilde{\Delta}^{1/3}T^{2/3}, \widetilde{\Delta}^{1/2}T^{5/8} \right\} + c_2^{3/2}\check{\Delta}^{2/5}T^{3/5} \right) & \alpha = -3 \end{cases}$$

## E BASE ALGORITHMS SATISFYING ASSUMPTION 1

In table 2 we summarize the results of this section.

Table 2: Parameters of the Base Algorithms. In this table we only show the magnitude of parameter, with $\widetilde{O}(\cdot)$ omitted except for the $\alpha$ column.

| Types of Games | $c_1$ | $\alpha$ | $c_1^\Delta$ |
|---|---|---|---|
| Zero-sum (NE) | $A + B$ | $-2$ | $1$ |
| General-sum (CCE) | $A_{\max}$ | $-2$ | $1$ |
| General-sum (CE) | $A_{\max}^2$ | $-2$ | $1$ |
| Potential (NE) | $m^2 A_{\max}$ | $-3$ | $1$ |
| Congestion (NE) | $m^2 F^3$ | $-2$ | $mF$ |
| Zero-sum Markov (NE) | $H^5 S(A + B)$ | $-2$ | $H^2$ |
| General-sum Markov (CCE) | $H^6 S^2 A_{\max}$ | $-2$ | $HS$ |
| General-sum Markov (CE) | $H^6 S^2 A_{\max}^2$ | $-2$ | $HS$ |
| Markov Potential (NE) | $m^2 H^4 S A_{\max}$ | $-3$ | $H^2$ |

### E.1 TWO-PLAYER ZERO-SUM MATRIX GAMES (NE)

In this part we consider the following algorithm: each player independently runs an optimal adversarial multi-armed bandit algorithm (e.g. EXP.3) and finally output the product of respective average policies of the whole time horizon. We will prove that this algorithm satisfies Assumption 1 in terms of learning NE in two-player zero-sum matrix games.

*Proof.* We adopt some new notations in this proof. Let $R^t \in [0,1]^{A \times B}$ be the reward matrix at episode $t$. The policy of the max and min players are represented by $x^t \in [0,1]^A, y^t \in [0,1]^B$. Each entry represents the probability they choose the corresponding action. The reward received by the max and min players are respectively ${x^t}^\top R^t y^t$ and $-{x^t}^\top R^t y^t$. With probability $1 - \delta$ the adversarial MAB algorithms satisfy

$$\frac{1}{T}\sum_{t=1}^T {x^t}^\top R^t y^t - \min_y \frac{1}{T}\sum_{t=1}^T {x^t}^\top R^t y \le c_{\text{adv}}\sqrt{AT}$$

$$\max_x \frac{1}{T}\sum_{t=1}^T x^\top R^t y^t - \frac{1}{T}\sum_{t=1}^T {x^t}^\top R^t y^t \le c_{\text{adv}}\sqrt{BT}$$

where $c_{\text{adv}} = \widetilde{O}(1)$. The output policy $\overline{x} = \sum_{t=1}^T x^t/T$ and $\overline{y} = \sum_{t=1}^T y^t/T$ satisfy

$$V_{\max}^{M^T}(\dagger, \overline{y}) + V_{\min}^{M^T}(\overline{x}, \dagger) = \max_x x^\top R^T \overline{y} + \min_y \overline{x}^\top R^T y \le c_{\text{adv}}\sqrt{BT} + \Delta + c_{\text{adv}}\sqrt{AT} + \Delta$$

By the definition of zero-sum game

$$\text{NEGAP}(\overline{x}, \overline{y}) \le \frac{V_{\max}^{M^T}(\dagger, \overline{y}) - V_{\max}^{M^T}(\overline{x}, \overline{y}) + V_{\min}^{M^T}(\overline{x}, \dagger) - V_{\min}^{M^T}(\overline{x}, \overline{y})}{2}$$

$$= \frac{V_{\max}^{M^T}(\dagger, \overline{y}) + V_{\min}^{M^T}(\overline{x}, \dagger)}{2} = \widetilde{O}\left(\sqrt{(A+B)T}\right) + 2\Delta.$$

Hence this algorithm satisfies Assumption 1 with $C_1(\epsilon) = \widetilde{O}\left((A+B)\epsilon^{-2}\right), c_1^\Delta = 2$. □

### E.2 MULTI-PLAYER GENERAL-SUM MATRIX GAMES (CCE)

In this part we consider the following algorithm: each player independently runs an optimal adversarial multi-armed bandit algorithm (e.g. EXP.3) and finally output the average joint policy of the whole time horizon. We will prove that this algorithm satisfies Assumption 1 in terms of learning CCE in multi-player general-sum matrix games.

*Proof.* We define the loss of player $i$ at episode $t$ by playing $a_i$ as

$$l_i^t(a_i) = 1 - \mathbb{E}_{a_{-i} \sim \pi_{-i}^t}\left[r_i(a_i, a_{-i}) \mid M^t\right]$$

then with probability $1 - \delta$, the adversarial MAB algorithm satisfies

$$\sum_{t=1}^{T} \langle \pi^t(\cdot), l_i^t(\cdot) \rangle - \min_{a_i \in \mathcal{A}_i} \sum_{t=1}^{T} l_i^t(a_i) \leq c_{\text{adv}} \sqrt{A_i T}, \quad c_{\text{adv}} = \widetilde{O}(1)$$

For convenience, we denote the reward function at timestep $t$ by $r^t$. Let the output policy $\pi = \sum_{t=1}^{T} \pi^t / T$, we have

$$V_i^{M^T}(\pi)$$

$$= \mathbb{E}_{\mathbf{a} \sim \pi} \left[ r_i^T(\mathbf{a}) \right] = \frac{1}{T} \sum_{t=1}^{T} \mathbb{E}_{a_i \sim \pi_i^t} \mathbb{E}_{a_{-i} \sim \pi_{-i}^t} \left[ r_i^T(a_i, a_{-i}) \right]$$

$$= 1 - \frac{1}{T} \sum_{t=1}^{T} \mathbb{E}_{a_i \sim \pi_i^t} \left[ l_i^t(a_i) \right] + \frac{1}{T} \sum_{t=1}^{T} \mathbb{E}_{a_i \sim \pi_i^t} \mathbb{E}_{a_{-i} \sim \pi_{-i}^t} \left[ r_i^T(a_i, a_{-i}) - r_i^t(a_i, a_{-i}) \right]$$

$$\geq 1 - \frac{1}{T} \min_{a_i \in \mathcal{A}_i} \sum_{t=1}^{T} l_i^t(a_i) - c_{\text{adv}} \sqrt{A_i/T} + \frac{1}{T} \sum_{t=1}^{T} \mathbb{E}_{a_i \sim \pi_i^t} \mathbb{E}_{a_{-i} \sim \pi_{-i}^t} \left[ r_i^T(a_i, a_{-i}) - r_i^t(a_i, a_{-i}) \right]$$

$$\geq \frac{1}{T} \max_{a_i \in \mathcal{A}_i} \sum_{t=1}^{T} \mathbb{E}_{a_{-i} \sim \pi_{-i}^t} \left[ r_i^t(a_i, a_{-i}) \right] - c_{\text{adv}} \sqrt{A_i/T} - \Delta$$

$$\geq \frac{1}{T} \max_{a_i \in \mathcal{A}_i} \sum_{t=1}^{T} \mathbb{E}_{a_{-i} \sim \pi_{-i}^t} \left[ r_i^T(a_i, a_{-i}) \right] - \Delta - c_{\text{adv}} \sqrt{A_i/T} - \Delta$$

$$\geq V_i^{M^T}(\dagger, \pi_{-i}) - c_{\text{adv}} \sqrt{A_i/T} - 2\Delta$$

By definition of CCE we know this algorithm satisfies Assumption 1 with $C_1(\epsilon) = \widetilde{O}\left(A_{\max}\epsilon^{-2}\right), c_1^\Delta = 2$ □

### E.3 MULTI-PLAYER GENERAL-SUM MATRIX GAMES (CE)

This part is very similar to the last part. Instead of using standard adversarial bandit algorithms, we use no-swap-regret algorithm for adversarial bandits (for example, Ito (2020)) and the proof is almost the same. We can achieve with probability $1 - \delta$,

$$\sum_{t=1}^{T} \langle \pi^t(\cdot), l_i^t(\cdot) \rangle - \min_{\psi_i} \sum_{t=1}^{T} \langle (\psi_i \diamond \pi^t)(\cdot), l_i^t(\cdot) \rangle \leq c_{\text{adv}} A_i \sqrt{T}, \quad c_{\text{adv}} = \widetilde{O}(1)$$

where $\psi_i$ is a strategy modification. By substituting all $\min, \max$ related terms correspondingly we get the proof for CE and $C_1(\epsilon) = \widetilde{O}\left(A_{\max}^2\epsilon^{-2}\right)$

### E.4 CONGESTION GAMES (NE)

In this part we will show the Nash-UCB algorithm proposed in Cui et al. (2022) satisfies Assumption 1. We carry out the proof by pointing out the modifications we need to make in their proof. In their proof, $k$ stands for the episode index instead of $t$ and $K$ is the total episodes instead of $T$.

**Lemma 10.** *(Modified Lemma 3 in Cui et al. (2022)) With high probability,*

$$\left| \tilde{r}_i^k - r_i \right|(\mathbf{a}) \leq \max_{i \in [m]} \|A_i(\mathbf{a})\|_{(V^k)^{-1}} \sqrt{\widetilde{\beta_k}}, \widetilde{\beta_k} = O(mF + Km\Delta^2)$$

*Proof.* We denote the average reward vector by $\bar{\theta}$ and the reward vector of the last epsiode by $\theta^T$, other notations are similar, then

$$\left| \tilde{r}_i^k - r_i^T \right|(\mathbf{a}) \leq \|A_i(\mathbf{a})\|_{(V^k)^{-1}} \left\| \hat{\theta} - \theta^T \right\|_{V^k}$$

$$\leq \|A_i(\mathbf{a})\|_{(V^k)^{-1}} \left( \left\| \hat{\theta} - \bar{\theta} \right\|_{V^k} + \left\| \bar{\theta} - \theta^T \right\|_{V^k} \right)$$

$$\leq \|A_i(\mathbf{a})\|_{(V^k)^{-1}} \left( \|\overline{\theta}\|_2 + \sqrt{\log \det \overline{V^k + \tilde{\iota}}} + \sqrt{Km}\Delta \right)$$

$\square$

The rest of the proof is carried out with the new $\widetilde{\beta}_k$ and finally the regret becomes

$$\text{Nash-Regret}(K) = \widetilde{O}\left(mF^{3/2}\sqrt{K} + mFK\Delta\right).$$

Finally this algorithm can be converted into a version with sample complexity guarantee and $C_1(\epsilon) = m^2F^3\epsilon^{-2}, c_1^\Delta = mF$ as stated in the original paper using the certified policy trick from Bai et al. (2020).

### E.5 MULTI-PLAYER GENERAL-SUM MARKOV GAMES (CCE,CE)

In this part we will show how to adapt the proof in Cui et al. (2023) to the non-stationary game case. For simplicity, we will follow the proof in Cui et al. (2023) in general and only point out critical changes. Note that they use $k$ as epoch index while we have been using $k$ as episode index. For consistency, we will use $\kappa$ as the episode index in this section. As a reminder, we will use $r^\kappa$, $P^\kappa$ and $M^\kappa$ to denote the reward function, the transition kernel and the game at episode $\kappa$.

We use the superscript $\kappa$ in $\mathbb{E}^\kappa[\cdot]$ to denote that the underlying game is $M^\kappa$. We further use $\kappa_h^k(j;s)$ to denote the episode index when state $s$ is visited for the $j$th time at step $h$ and epoch $k$ in the no-regret learning phase (Line 12 in Algorithm 3), and we use $\overline{\kappa}_h^k(j;s)$ to denote the episode index when state $s$ is visited for the $j$th time at step $h$ and epoch $k$ in the no-regret learning phase (Line 12 in Algorithm 3). We will change the algorithm in Line 34 where we replace $n_h^k(s_h)$ with $\sum_{l=1}^{k-1} T_h^k(s_h)$. We will modify all the lemmas in the proof below. We use $N^k$ to denote $\sum_{l=1}^k n^k$.

First, we will replace $\mathbb{E}_{\mathbf{a}\sim\pi_h^{k,t_h^k(j;s)}(\cdot|s)}[\cdot]$ with $\mathbb{E}^{\kappa_h^k(j;s)}_{\mathbf{a}\sim\pi_h^{k,t_h^k(j;s)}(\cdot|s)}[\cdot]$ in all the lemmas, which takes the expectation with the underlying game when $\pi_h^{k,t_h^k(j;s)}(\cdot \mid s)$ is used. It is easy to verify that Lemma 35, Lemma 36, Lemma 37 hold after the modification.

Second, we will replace $n_h^k(s)$ with $\sum_{l=1}^{k-1} T_h^l(s)$ and $n^k d_h^{\pi^k}(s)$ with $\sum_{J=1}^{n^k} d_h^{\pi^k}(s;k,J)$, where $d_h^{\pi^l}(s;k,J)$ is the visiting density for model at epoch $k$ and $J$th trajectory sampled in the policy cover update phase. In addition, we also add the following argument in the lemma:

$$n_h^k(s) \vee \text{Trig} \geq \frac{1}{2}\left(\sum_{l=1}^{k-1} \frac{n^l}{N^{k-1}} \sum_{j=1}^{N^{k-1}} d_h^{\pi^l}(s;k,j)\right) \vee T_{\text{Trig}},$$

where $d_h^{\pi^l}(s;k,j)$ is the visiting density for model at epoch $k$ and $j$th trajectory sampled in the no-regret learning phase. It is easy to verify that Lemma 38 hold after the modification.

Third, we will consider a baseline model $M^0$, which can be the game at any episode, and use $V_{h,i}^\pi(s)$ to denote the corresponding value function. Now we show that Lemma 39, Lemma 40 and Lemma 41 holds with an addition tolerance $\Delta$.

**Lemma 11.** *(Modified Lemma 39 in Cui et al. (2023)) Under the good event $\mathcal{G}$, for all $k \in [K]$, $h \in [H]$, $i \in [m]$, $s \in \mathcal{S}$, we have*

$$\overline{V}_{h,i}^k(s) \geq V_{h,i}^{\dagger,\pi_{-i}^k}(s) - \sum_{h'=h}^H \Delta_h.$$

*Proof.* Note that we have

$$\left|\mathbb{E}^{\kappa_h^k(j;s)}_{\mathbf{a}\sim\pi_h^{k,t_h^k(j;s)}(\cdot|s)}\left[r_{h,i}(s,\mathbf{a}) + \overline{V}_{h+1,i}^k(s')\right] - \mathbb{E}^{M^0}_{\mathbf{a}\sim\pi_h^{k,t_h^k(j;s)}(\cdot|s)}\left[r_{h,i}(s,\mathbf{a}) + \overline{V}_{h+1,i}^k(s')\right]\right| \leq \Delta_h.$$

The rest of the proof follows Cui et al. (2023). $\square$

**Lemma 12.** *(Modified Lemma 40 in* Cui et al. (2023)*) Under the good event $\mathcal{G}$, for all $k \in [K]$, $h \in [H]$, $i \in [m]$, $s \in \mathcal{S}$, we have*

$$\underline{V}^k_{h,i}(s) \leq V^{\pi^k}_{h,i}(s) + \sum_{h'=h}^{H} \Delta_h.$$

*Proof.* The proof follows the proof for Lemma 11. □

**Lemma 13.** *(Modified Lemma 41 in* Cui et al. (2023)*) Under the good event $\mathcal{G}$, for all $k \in [K]$, $i \in [m]$, we have*

$$\overline{V}^k_{1,i}(s_1) - \underline{V}^k_{1,i}(s_1) \leq \widetilde{O}\left(\mathbb{E}^{M^o}_{\pi^k}\left[\sum_{h=1}^{H}\sqrt{\frac{H^2 A_i T_{\mathrm{Trig}}}{n^k_h(s_h) \vee T_{\mathrm{Trig}}}}\right]\right) + 2\Delta.$$

*Proof.* The proof follows the proof for Lemma 11. □

**Lemma 14.** *(Modified Lemma 42 in* Cui et al. (2023)*) Under the good event $\mathcal{G}$, for all $i \in [m]$, we have*

$$\sum_{k=1}^{K} n^k \max_{i \in [m]}\left(\overline{V}^k_{1,i}(s_1) - \underline{V}^{\pi^k}_{1,i}(s_1)\right) \leq \widetilde{O}\left(H^2\sqrt{SA_{\max}T_{\mathrm{Trig}}N}\right).$$

*Proof.* By Lemma 13 and the proof in Cui et al. (2023), we only need to bound $\sum_{k=1}^{K} n^k \mathbb{E}^{M^o}_{\pi^k}\sqrt{\frac{1}{n^k_h(s_h) \vee T_{\mathrm{Trig}}}}$. By the definition of $\Delta$, we can easily prove that

$$\sum_{s \in \mathcal{S}}\left|\frac{n^k}{N^k}\sum_{j=1}^{N^k} d^{\pi^k}_h(s;k+1,j) - \sum_{J=1}^{n^k} d^{\pi^k}_h(s;k,J)\right| \leq n^k \Delta,$$

$$\sum_{s \in \mathcal{S}}\left|n^k d^{\pi^k}_h(s) - \left(\sum_{l=1}^{k}\frac{n^l}{N^k}\sum_{j=1}^{N^k} d^{\pi^l}_h(s;k+1,j) - \sum_{l=1}^{k-1}\frac{n^l}{N^{k-1}}\sum_{j=1}^{N^{k-1}} d^{\pi^l}_h(s;k,j)\right)\right| \leq N^k \Delta.$$

and we have

$$\sum_{s \in \mathcal{S}}\left(\sum_{l=1}^{k}\frac{n^l}{N^k}\sum_{j=1}^{N^k} d^{\pi^l}_h(s;k+1,j) - \sum_{l=1}^{k-1}\frac{n^l}{N^{k-1}}\sum_{j=1}^{N^{k-1}} d^{\pi^l}_h(s;k,j)\right) - 2\sum_{l=1}^{k-1}\frac{n^l}{N^{k-1}}\sum_{j=1}^{N^{k-1}} d^{\pi^l}_h(s;k,j) \leq 4N^k \Delta.$$

Then we have

$$\sum_{k=1}^{K} n^k \mathbb{E}^{M^o}_{\pi^k}\sqrt{\frac{1}{n^k_h(s_h) \vee T_{\mathrm{Trig}}}}$$

$$= \sum_{k=1}^{K} n^k \sum_{s \in \mathcal{S}} d^{\pi^k}_h(s)\sqrt{\frac{1}{n^k_h(s) \vee T_{\mathrm{Trig}}}}$$

$$\leq \sum_{s \in \mathcal{S}}\sum_{k=1}^{K} n^k d^{\pi^k}_h(s)\sqrt{\frac{2}{(\sum_{l=1}^{k-1}\frac{n^l}{N^{k-1}}\sum_{j=1}^{N^{k-1}} d^{\pi^l}_h(s;k,j)) \vee T_{\mathrm{Trig}}}} \quad \text{(Lemma 38 in Cui et al. (2023))}$$

$$\leq NK\Delta + \sum_{s \in \mathcal{S}}\sum_{k=1}^{K}\left(\sum_{l=1}^{k}\frac{n^l}{N^k}\sum_{j=1}^{N^k} d^{\pi^l}_h(s;k+1,j) - \sum_{l=1}^{k-1}\frac{n^l}{N^{k-1}}\sum_{j=1}^{N^{k-1}} d^{\pi^l}_h(s;k,j)\right)$$

$$\sqrt{\frac{2}{(\sum_{l=1}^{k-1}\frac{n^l}{N^{k-1}}\sum_{j=1}^{N^{k-1}} d^{\pi^l}_h(s;k,j)) \vee T_{\mathrm{Trig}}}}$$

$$\leq 2NK\Delta + \sum_{s \in \mathcal{S}}\sqrt{32\sum_{l=1}^{K}\frac{n^l}{N^K}\sum_{j=1}^{N^K} d^{\pi^l}_h(s;K,j)} \quad \text{(Lemma 38 and Lemma 53 in Cui et al. (2023))}$$

$$\leq 2NK\Delta + \sqrt{32SN}.$$

$\square$

Lemma 43 in Cui et al. (2023) holds directly with the modified update rule. As a result, following Theorem 4 in Cui et al. (2023), the same sample complexity result holds for learning an $\epsilon + \widetilde{O}(HS\Delta)$-CCE. Hence $C_1(\epsilon) = H^6 S^2 A_{\max} \epsilon^{-2}, c_1^\Delta = HS$.

### E.6 MARKOV POTENTIAL GAMES (NE)

This setting is rather straightforward. Algorithm 3 in Song et al. (2021) serves as a base algorithm. By noticing that any weighted average of the samples of rewards shifts by no more than $O(\Delta)$ in the non-stationary environment and by the very similar argument we made in Lemma 3 or proof of Theorem 1 in Mao et al. (2021) we can see $C_1(\epsilon) = m^2 H^4 S A_{\max} \epsilon^{-3}, c_1^\Delta = O(H^2)$.

