# OpenReview forum: "A Black-box Approach for Non-stationary Multi-agent Reinforcement Learning"
_ICLR.cc/2024/Conference — ICLR 2024 poster_

### Official Review · Reviewer_nAA2 · 2023-10-28

**Soundness:** 3 good
**Presentation:** 3 good
**Contribution:** 3 good
**Rating:** 6
**Confidence:** 3

**Summary:**

This work points out two main challenges in generalizing single-agent non-stationary algorithms
to non-stationary Markov games: (1) applying the online learning techniques to tackle non-stationarity In the bandit feedback setting is problematic; (2) equilibria is multi-agent games are not unique. To overcome these obstacles, the authors propose a black-box approach to transform algorithms designed for stationary games into ones capable of handling non-stationarity.Also, regret bounds in different settings are provided.

**Strengths:**

Regarding the originality, the studied problem is new (though there has been some works on single-agent non-stationary settings). Also, it is obvious that this problem contains its intrinsic difficulties, so I would agree this study is resolving a significant problem.

For clarity, this work is well-presented though it still can be improved. It is easy to understand its motivation of handling the challenges of non-stationary games. Introducing the the case of known non-stationary budget is very helpful to understand the main results.

**Weaknesses:**

Lack of discussions on the definitions, assumptions, and results. For exmaple, from Definition 4, if the non-stationarity degree of Markov game is zero, should this setting degenerates to the traditional stationary setting? It seems very straightforward, but I prefer to get confirmed from the paper. I put more comments in the questions section.

The introduction section contains many terms that have not been clearly defined, such as non-stationarity budget and black-box approach. I would prefer to see a revision on the introduction section including providing a high-level introduction to these concepts and giving some references on real-world scenarios where we really need this algorithm.

**Questions:**

1. If the non-stationarity degree of Markov game is zero, should this setting degenerates to the traditional stationary setting?
2. Also, when $\Delta=0$, will the regret bounds (Corollary 1) degenerates to the regret bounds in the stationary case? Will it match the existing rate?
3. I am not familiar with Assumption 1 and Assumption 2. Are they also common to be made in stationary MAML works?
4. Continuing the questions about Assumption 1 and Assumption 2, when $\Delta$ is not zero, the learned EQ may have a constant distance $c^\Delta \Delta$ from the desired EQ. Is this true? Why should we learn this joint policy that has a constant gap to an EQ?
5. It is a purely theoretical work, but I am still interested in real-world scenarios where the proposed algorithm can find its applications. Any examples?

---

> ### Author Response · Authors · 2023-11-15
> **Response to Reviewer nAA2**
>
> Thank you very much for your careful review and constructive suggestions! Please find our response to your questions and concerns below:
> - **Degenerates to the traditional stationary setting?** Yes, it degenerates to the traditional stationary setting when the non-stationarity degree of Markov game is zero. We have incorporated this point to the revised version.
> - Yes, the bound degenerates to the bound in the stationary case. However, it does not match the existing rate. When the environment is stationary, the algorithm is the vanilla Explore-then-Commit algorithm, which is suboptimal. We have incorporated this point to the revised version.
> - **Assumption 1, 2:** In stationary settings, i.e. $\Delta=0$, Assumption 1, 3 degenerate to the ordinary sample complexity assumptions, which is common in machine learning. Assumption 2 is not often made in literature, but we construct Protocol 1 (Algorithm 2 in the revised version) to convert an algorithm satisfying Assumption 3 to an algorithm satisfying Assumption 2. You are correct that the learned policy may have $c^\Delta\Delta$ gap from the equilibrium. We do not intend to inject such gap, but this is the best we can hope for algorithms designed for stationary environments. If we had base algorithms that can achieve arbitrary small $\epsilon$ gap, we would not need to devise a new algorithm for the non-stationary environment.
> - **Motivating Examples:** Non-stationarity is ubiquitous in multi-agent systems, here are some concrete examples. First, consider traffic routing in a road network, each driver tries to get to their destination as soon as possible. Due to congestion, the action of one player can affect others' payoffs. The underlying environment can be changing due to weather conditions, pedestrian conditions and road policy changes. Second, consider the financial market. Stock markets, currency exchange, and other financial markets can be highly non-stationary due to constantly changing economic conditions, news events, and regulations. Traders need to adapt their investment strategies to stay profitable. Third, consider professional sports teams, where each to them adopt a strategy to play against other teams throughout a season. In different seasons, game rules, player rosters and the performance of each player can all be different, introducing non-stationarity. In particular, at the end of Section 1.1, we provide a possible application of adaptively tracking Nash Equilibria in the first example.
> - **Introduction:** We add real-world multi-agent scenarios where non-stationarity is important in the revised version with references. We added an explicit definition of black-box algorithm. We also highlight the sentence defining nonstationarity budget in the Introduction. If there is any concept you think should be clarified, please tell us and we are happy to add accordingly.

---

> > ### Comment · Reviewer_nAA2 · 2023-11-17
> >
> > Thanks for your responses! I am feeling this paper has been much more clear. And my concerns on the assumptions and the constant gap to desired equilibrium are well addressed. I would also agree with that those given motivation examples regarding games could be highly nonstationary. So, I would like to increase my scores to 6.

---

> > > ### Author Response · Authors · 2023-11-17
> > > **Response to Reviewer nAA2**
> > >
> > > Thank you for your positive support! If you have any further questions, please feel free to ask.

---

### Official Review · Reviewer_AT29 · 2023-10-29

**Soundness:** 3 good
**Presentation:** 2 fair
**Contribution:** 3 good
**Rating:** 6
**Confidence:** 2

**Summary:**

This paper delves into the multi-agent general sum Markov games framework within nonstationary environments. The scenario involves m agents where, at each step, a policy selects a joint action (a_h = (a_1, ..., a_m)), and each player i receives a random reward (r_{h,i}). The subsequent state at h+1 (s_{h+1}) is drawn from P_h( |s_h, a_h). To accommodate nonstationarity, Markov games can undergo changes in transition probability or reward function over time.

The primary contribution of the authors lies in highlighting the nontrivial nature of generalizing an algorithm for nonstationary Markov games to a parameter-free version due to distinct objectives. They advocate a black box approach for nonstationary games and subsequently analyze the algorithm for both known and unknown nonstationary budgets.

For the purposes of this study, the authors define epsilon-approximate Nash equilibrium, coarse correlated equilibrium, and correlated equilibrium using the gap between the value function given a policy. Regret over T is then defined as the summation of the gap given a policy.

The proposed method initially considers black box oracles capable of learning and testing equilibrium in near-stationary environments. The learning equilibrium oracle requires the fulfillment of assumption 1, where access to an oracle outputting a good EQ policy within C_1 samples is available. The test equilibrium oracle assumes access to an oracle that outputs False when a policy is not a good EQ policy and True when the policy is good EQ.

In the warm-up section, the authors present a MARL algorithm (Algorithm 1) for nonstationary environments with a known nonstationary budget. Regret bounds, depending on the bases, are achieved by setting rounds of each phase and the value of epsilon with a nonstationary budget, as detailed in Corollary 1.

In the subsequent section, the authors address the case of an unknown nonstationary budget. In this scenario, a testing oracle is required, which can be constructed from Protocol 1 using a learning oracle. They propose Algorithm 2, comprising learning EQ, testing EQ, and a meta-algorithm, achieving regret bounds as detailed in Theorem 1 and 2.

**Strengths:**

-This paper is the pioneer in addressing nonstationarity in Multi-Agent Reinforcement Learning concerning equilibrium.

-The proposed framework is designed as a black box, ensuring compatibility with various bases.

**Weaknesses:**

-Algorithm 2, as outlined in the paper, necessitates a test EQ algorithm to detect nonstationarity in the case of an unknown budget. The assumption of the existence of a testing EQ trained offline appears to be a strong assumption in the context of online learning problems.

-The tightness of regret is not clear.  For example, it is not evident whether the results achieved in this paper surpass the Bandit over Bandit approach with a sliding window multi-agent RL algorithm.



minor:
It would enhance clarity to provide an explanation for the definition of '(\Delta)-EQ' in Assumptions.

**Questions:**

-It seems that the regret from training the test EQ is not factored into the results presented in Theorems 1 and 2. What implications might there be if the training cost is considered in the regret?

-Despite the authors discussing the inapplicability of the Bandit over RL approach to this problem, what if we were to consider a bandit over (sliding window) MARL approach for addressing nonstationary MARL? Due to the order of the regret in Theorems 1 and 2, it remains unclear whether the suggested method is superior to the bandit over bandit approach.

---

> ### Author Response · Authors · 2023-11-15
> **Response to Reviewer AT29**
>
> Thank you very much for your careful review!
> - **TEST_EQ:** The regret from TEST_EQ is counted into the total regret. **The whole algorithm is run online and no offline training is needed.** In particular, this part of regret is calculated at the very beginning of the proof for Lemma 8 in the Appendix.
> - **Tightness:** We did consider constructing a lower bound for non-stationary games, but that seems too difficult. While the lower bound for non-stationary bandits are known for a long time, there is no non-trivial regret lower bounds for stationary normal form games with bandit feedback to the best of our knowledge. Hence it is too optimistic to hope for a regret lower bound for our setting directly.
> - **Bandit over MARL:** **We discussed the challenges of bandit over MARL in the last paragraph of Section 3 and a more detailed explanation is provided in Appendix B.2.** A quick explanation is as follows. In games different actions are competing with different best response actions whereas in bandits all arms are competing with the single best arm. In games, comparing the value of two actions does not tell us which is better because the gap can be calculated toward different best responses. The high-level idea of Bandit over Bandit is to divide the whole time horizon into several segments, run an algorithm with known non-stationary budget in each of them and run an adversarial bandit algorithm on the segments. A crucial part is to use the sum of rewards in a segement as the reward for the adversarial bandit algorithm. The problem in the multi-agent setting is that the sum of rewards is meaningless for evaluating the performance as explained before. One may refer to the Appendix for a more formal explanation with formulae.
> - **$\epsilon$-EQ:** The definitions are in Definition 1,2,3 and the subsequent paragraph.

---

> > ### Comment · Reviewer_AT29 · 2023-11-17
> >
> > Thank you for your reply. I comprehend that the test_EQ contributes to the overall regret.
> > I still have an inquiry regarding whether bandit over MARL works or not. Let $\pi(w)$ be a sliding window MARL policy with $w$ sliding window. Let $w^*$ be near the optimal sliding window and $w_t$ is a chosen window from adversarial bandits at time $t$. Then $\sum_{t=1}^T (V_i^M(\dagger,\pi_{-i})-V_i^M(\pi))=[\sum_{i=1}^T V_i^M(\dagger,\pi_{-i})-\sum_{h=1}^{T/H}V_i^M(\pi(w^*))]+[\sum_{h=1}^{T/H}V_i^M(\pi(w^*))-\sum_{t=1}^T V_i^M(\pi(w_t))]$. I still could not find why the sum of the value function is meaningless. This is because the second term seems to be bounded using EXP3 regret analysis. Please inform me if there is any aspect I may be overlooking.

---

> ### Author Response · Authors · 2023-11-17
> **Response to Reviewer AT29**
>
> Thank you very much for your reply and carefully examining the details. To explain the Bandit over MARL approach in detail, we further expand the notation. Let $\pi_t$ be the policy that the agent take by using the Bandit over MARL algorithm where $t\in[T]$. Let $\pi_t^*$ be the policy that the agent would take by committing to the best parameter $w^*$ in a segment, $t\in[H]$. The first term in the expansion of regret can be written as $$\sum_{t=1}^TV_i^M(\dagger, \pi_{t,-i})-\sum_{h=1}^H\sum_{t=1}^HV_i^M(\pi^*_t).$$ Since $\pi^*_t$ is competing with its best response, which may not equal the best response of $\pi_t$, this term cannot be bounded by regret guarantee of the base algorithm. In contrast, in the bandit setting, no matter what the policy is, it is always comparing with $a_t^*$, the optimal arm at timestep $t$. Hence in that scenario the corresponding term can be bounded with the regret of the base algorithm.

---

> > ### Comment · Reviewer_AT29 · 2023-11-19
> >
> > I understand that opting for bandit over MARL may not be suitable for this context. With most of my concerns addressed, despite the potential suboptimal tightness of the regret bounds, I see the contribution of this paper as significant, especially as the initial step towards nonstationary MARL. As a result, I've decided to increase my score to 6.

---

> > > ### Author Response · Authors · 2023-11-19
> > > **Response to Reviewer AT29**
> > >
> > > Thank you for your positive support! If you have any further questions, please feel free to ask.

---

### Official Review · Reviewer_FQRF · 2023-11-01

**Soundness:** 3 good
**Presentation:** 2 fair
**Contribution:** 2 fair
**Rating:** 6
**Confidence:** 2

**Summary:**

The authors study the problem of multi-agent reinforcement learning with only bandit feedback and where the underlying game may be non-stationary---that is, slightly changing at each iteration of the learning process. The authors propose to quantify the degree to which the game changes at each iteration using a total variation-inspired quantity. They show that since standard algorithms already provide regret bounds that scale reasonably even under non-stationarity bounded by this TV quantity, one can simply use the algorithms and creatively reset them whenever the non-stationarity has become too great (since between resets, the algorithms are expected to already handle the bounded amount of non-stationarity). The main technical ingredients the authors use is 1) a testing schedule that efficiently tests for non-stationarity to determine when resets should happen and 2) algorithms that test whether joint strategies constitute equilibria by reducing to single-agent learning problems.

**Strengths:**

The paper studies a reasonable and easily stated problem: how hard is multi-agent RL when your training environment is slightly changing at each round. It is (for the most part) well-written and makes its technical contributions clear. The quantity that the authors proposed measuring their regret bounds by, total variation non-stationarity budget $\Delta$, is a reasonable analogue of total variation distance for this application. As a fairly pessimistic quantity, the claimed bounds' dependences on $\Delta$ also seem reasonable. The proposed algorithms are intuitive and admit a fairly clean analysis, although I have a hard time evaluating their significance, which I'm hoping the authors could clarify through my questions.

**Weaknesses:**

* The claimed regret bounds in Table 1 seem somewhat coarse given that the proposed notion of "total variation non-stationary budget" $\Delta$ scales linearly in $H \times S \times A$. Taking that into account, all of the regret bounds for multi-round games displayed in Table 1 look like they should scale linearly or superlinearly in horizon length and linearly in state size for general-sum Markov games and the square-root of state-size for zero-sum games. It seems one could get a similar regret bound (maybe even of order $O(\sqrt{T})$) by just flattening each game into a single round game with $H \times S \times A$ actions and having each player run Exp3 on those actions? The results in Table 1 could still be really interesting, but its difficult to put them into perspective because their main feature is $\Delta$, which the authors seem to be proposing the use of. It would be helpful if the author could clarify if $\Delta$ is already a commonly used quantity that is known to be small in practice.
* I'm not familiar with restarting-based algorithms so I find it harder to evaluate the novelty/significance of the results in 5.2 and 5.3. However, the contributions listed in 5.1, namely the reduction of equilibria testing to single agent learning, are---to the best of my knowledge---already known (in the case of CCE testing) or very direct in the sense that its a one-two sentence reduction and the first thing one would try (in the case of CE).
* The notation in the paper is a bit difficult to read in certain cases. In particular, the use of $\Delta$ in exponents could be clarified. (See below)

**Questions:**

* Can you clarify the difference between $c_1^\Delta$ versus $c_1$ and $c_2^\Delta$ versus $c_2$? Is $c_1^\Delta$ referring to $c_1$ to the power $\Delta$, or is it some other constant entirely?
* Is your definition of a "total variation non-stationarity budget" $\Delta$ used in any other works? As mentioned previously, it seems like a very pessimistic quantity (as one would expect from a TV-like quantity) scaling linearly in # of states * # of players * # of actions * horizon.
* Why do online learning algorithms fail in this setting because of bandit feedback? Its not obvious why bandit/semi-bandit algorithms that give regret bounds even for adaptive adversaries are not applicable in this setting. In fact, a bounded total variation non-stationarity budget seems like it would also imply that the second-order regret bounds of online learning algorithms should be small in this setting.
* Are the first 5 columns of Table 1 referring to single-timestep games?
* Is there a reason the terminology alternates between "Protocol" and "Algorithm"?
* Typo on page 6: chanllanges -> challenges, page 5: For simplicity, We -> For simplicity, we.

---

> ### Author Response · Authors · 2023-11-15
> **Response to Reviewer FQRF**
>
> Thank you very much for your careful review and constructive suggestions! Please find our response to your questions and concerns below:
> - **$\Delta$**: We apologize for the confusion. $c_1^\Delta,c_2^\Delta,c_3^\Delta$ are constants depending on the base algorithms (not $c$ to the power $\Delta$). The purpose of writing in this way is to remind readers that it is the constant coefficient of $\Delta$ in the assumptions and to keep the notation concise.
> - **Total Variation:** We apologize for our typos in the paper. Following prior works, the nonstationarity takes the max over states and actions instead of taking sum. Similar notations are common in literature. For instance, the measure Wei & Luo (2021) use for single-agent RL is $$\Delta_{[t,t+1]}=H\sum_{h=1}^H\max_{s,a}\left|r_h^t(s,a)-r_h^{t+1}(s,a)\right|+H^2\sum_{h=1}^H\max_{s,a}\left\|r_h^t(s,a)-r_h^{t+1}(s,a)\right\|$$ which has an $O(H^2)$ difference in order from our measure.
> - **Flattening the Game**: Could you be more specific on how to flatten the game? If you mean treating each single player pure strategy as a bandit arm, the number would be exponential. At each timestep, the number of different pure-strategy policies is $A^S$ since we may choose $A$ different actions for each state. **The number of different combinations of single-step policies is hence $A^{SH}$, which is exponential**. Besides, each player running EXP.3 can achieve $O\left(\sqrt{T}\right)$ individual regret, but **there is no NE/CE/CCE-regret guarantee in the stationary game setting to the best of our knowledge**. Hence it is open to achieve $O\left(\sqrt{T}\right)$ regret in the harder non-stationary case using EXP.3.
> - **Bandit Algorithms**: Directly applying (non-stationary) bandit algorithms to our setting does not work because a (second order) regret bound for single player **does not imply a (second order) regret bound in the multi-player setting**. In essence, this is because the **changing policy of other players provide one extra source of non-stationarity in the view of each player**. Many prior works, including both theoretical and empirical ones, have touched on this challenge (Cui et al. 2023, Lowe et al. 2017)
> - **Novelty**: In the non-stationary bandit literature, testing algorithms are usually used. While constructing TEST_EQ is relatively straightforward, we think it is important to state it rigorously and prove the correctness. This lays the foundation for future work that need testing algorithms in non-stationary games. This construction is a side contribution of this paper while the main technical contributions lie in comprehensively analyzing the difficulties incurred by the game setting (Section 3) and designing the (parameter-free) learning algorithm with no-regret guarantee (Section 4,5). In particular, designing such algorithms require many detailed while non-trivial adaptations. We raise two examples here. First, the test scheduling in Section 5.2 is tailored for Assumption 2. The scheduling we use assures that with high probability TEST_EQ terminates after running for desired number of timesteps, otherwise TEST_EQ may not generate a meaningful output. In prior works, when overlapping of different tests happen, their lengths get cut. The second example is that we use a new criterion to partition the time horizon into near-stationary intervals as mentioned in Remark 2. This technique improves the bound we get.
> - **Table 1:** You are correct, the first five rows refer to single-timestep games.
> - There was no particular reason for alternating between 'Protocol' and 'Algorithm'. We used 'Algorithm' for the main algorithms and 'Protocol' for the auxiliary algorithms. We have changed them all into 'Algorithm' in the revised version.
> - We have fixed our typos.
>
> Please check the global response above for all modifications in the revised version.
> ```
> Chen-Yu Wei and Haipeng Luo. Non-stationary reinforcement learning without prior knowledge: an optimal black-box approach. 2021.
> Qiwen Cui, Kaiqing Zhang, and Simon S Du. Breaking the curse of multiagents in a large state space: RL in markov games with independent linear function approximation. 2023.
> Ryan Lowe, Yi Wu, Aviv Tamar, Jean Harb, Pieter Abbeel, Igor Mordatch. Multi-Agent Actor-Critic for Mixed Cooperative-Competitive Environments. 2017
> ```

---

> > ### Author Response · Authors · 2023-11-21
> > **Sincerely Looking Forward to Further Feedbacks**
> >
> > Dear Reviewer,
> >
> > Thank you for your time and efforts in reviewing our work. We have provided detailed clarification to address the issues raised in your comments. If our response has addressed your concerns, we would be grateful if you could re-evaluate our work.
> >
> > If you have any additional questions or comments, we would be happy to have further discussions.
> >
> > Thanks,
> >
> > The authors

---

> ### Comment · Reviewer_FQRF · 2023-11-23
> **Reviewer response**
>
> Thanks for the clarification! That $\Delta$ is closer to an infinity than 1 norm resolves my main concern regarding the blowup of the regret bounds. I've revised the original review accordingly.

---

> > ### Author Response · Authors · 2023-11-23
> > **Response to Reviewer FQRF**
> >
> > Thank you for your positive support! If you have any further questions, please feel free to ask.

---

### Official Review · Reviewer_Zou4 · 2023-11-01

**Soundness:** 3 good
**Presentation:** 3 good
**Contribution:** 3 good
**Rating:** 6
**Confidence:** 4

**Summary:**

The paper considers the multi-agent RL problem in non-stationary Markov games, as a generalization from the non-stationary bandit problem.

**Strengths:**

- The paper is well-written and provides a good summary of the existing work on multi-agent RL. The black-boxes approach not only makes the results of this paper more general, but also helps the reader to understand the approach to this problem at a high-level.

**Weaknesses:**

- The paper seems to be the first work that considers the non-stationary Markov games, but this particular game setup is not well-motivated, except it is a natural generalization from the non-stationary bandit problem. I expect the author to provide some discussion on the motivation of this game setup, possibly with some real-world application scenarios.
- Is Assumption 3 a reasonable assumption? I expect the author to add some discussion to justify this assumption (you did for Assumption 1, 2). I think only when \delta is small, there are existing algorithms that can satisfy this assumption.
- It seems to me that the key idea of the proposed algorithm is largely based on that of Wei & Luo (2021). I expect the author to provide some discussion/summary on the challenges of extending their approach to the game setup and highlight some key differences between the two algorithms.

**Questions:**

See my comments above

---

> ### Author Response · Authors · 2023-11-15
> **Response to Review Zou4 - Part I**
>
> Thank you very much for your careful review and constructive suggestions! Please find our response to your questions and concerns below:
> - **Motivating Examples:** Non-stationarity is ubiquitous in multi-agent systems, here are some concrete examples. First, consider traffic routing in a road network, each driver tries to get to their destination as soon as possible. Due to congestion, the action of one player can affect others' payoffs. The underlying environment can be changing due to weather conditions, pedestrain conditions and road policy changes. Second, consider the financial market. Stock markets, currency exchange, and other financial markets can be highly non-stationary due to constantly changing economic conditions, news events, and regulations. Traders need to adapt their investment strategies to stay profitable. Third, consider professional sports teams, where each of them adopts a strategy to play against other teams throughout a season. In different seasons, game rules, player rosters and the performance of each player can all be different, introducing non-stationarity. In particular, **at the end of Section 1.1**, we provide a potential application of adaptively tracking Nash Equilibria in the first example.
> - **Assumption 3:** Assumption 3 is a direct corollary of the assumption Wei & Luo (2021) made on the stationary bandit algorithms. **Details can be found in the proof of Proposition 3 in the appendix**. We restate Assumption 1 of Wei & Luo (2021) as follows. Let $f_t:\Pi\to[0,1]$ be the reward function at timestep $t$ where $\Pi$ is the policy set of the player and $t\in[T]$. At timestep $t$, $f^\star_t=\max_{\pi\in\Pi}f_t(\pi)$ is the optimal mean reward and $R_t$ is the mean reward received by the player. The algorithm outputs an auxiliary quantity $\widetilde{f}\_t$ at timestep $t$ satisfying that with probability $1-\delta/T$, for all timesteps $t$, for some function $\rho:[T]\to\mathbb{R}$, $$\widetilde{f}\_t\geq\min_{\tau\in[1,t]}f_\tau^\star-\Lambda_{[1,t]},\quad \frac{1}{t}\sum_{\tau=1}^t\left(\widetilde{f}\_\tau-R_\tau\right)\leq\rho(t)+\Lambda_{[1,t]}.$$ Here $\Lambda$ is the non-stationarity measure. Using the fact that $$\max_{\tau\in[1,t]}f_\tau^\star-\min_{\tau\in[1,t]}f_\tau^\star\leq\Lambda_{[1,t]}$$ we have that for any timestep $s\in[1,t]$, $$\frac{1}{t}\sum_{\tau=1}^t\left(\max_{\tau'\in[1,t]}f_{\tau'}^\star-R_\tau\right)\leq\rho(t)+3\Lambda_{[1,t]}.$$ For single-agent RL, Q-UCB algorithm achieves $\rho(t)=\widetilde{O}\left(\sqrt{H^5SA/t}+H^3SA/t\right)$ and $\Lambda_{1,t}=\widetilde{O}(H^2\Delta)$. Thus, the average policy of behavior policies in $[1,t]$ satisfies Assumption 3 with $C_3(\epsilon,\delta)=\widetilde{O}(H^5SA/\epsilon^2)$ and $c_3^\Delta=\widetilde{O}(H^2)$.
> - **Challenges extending MASTER to Games:** The challenges of extending non-stationary bandit algorithms to the game setup is **discussed extensively in Section 3**. The main challenge of extending MASTER to the game setup is as follows. The most crucial part of MASTER is to make use of the auxiliary quantity $\widetilde{f}\_t$ to design clever criteria that detect changes in the underlying environment that significantly affect the performance. Usually, $\widetilde{f}\_t$ is the optimistic value of the action. This quantity is easy to find for single-agent RL because many algorithms are UCB-based. However, MARL algorithms are more diverse. For instance, every agent running adversarial bandit algorithm converges to a CCE in general-sum Markov games, where such value is hard to extract. Even if we try to build an algorithm on top of a UCB-based algorithm, like Nash-UCB, it is hard to extend Assumption 1 in Wei & Luo (2021) to games. A quantity similar to $\min_{\tau\in[1,t]}f_\tau^\star$ is difficult to define because the best response is different for different policies and comparing mean rewards of best responses to different policies is pointless. This difficulty is also discussed in Section 3.
> ```
> Chen-Yu Wei and Haipeng Luo. Non-stationary reinforcement learning without prior knowledge: an optimal black-box approach. 2021.
> Peter Auer, Pratik Gajane, and Ronald Ortner. Adaptively tracking the best bandit arm with an unknown number of distribution changes. 2019.
> ```

---

> ### Author Response · Authors · 2023-11-15
> **Response to Reviewer Zou4 - Part II**
>
> - **Differences between MASTER and our algorithm:** First, we make sample complexity assumptions on the base algorithms. In contrast, MASTER assumes the existence of auxiliary quantity $\widetilde{f}_t$ and makes a more regret-like assumption. Second, we adopt an explore-commit-test paradigm. In contrast, MASTER automatically test for significant changes while running the learning algorithms, which is not easy to achieve for games as discussed in Section 3. Third, while sharing the same high-level idea of base algorithm, adaptation to sample-complexity-based assumption is made in our algorithm, as explained in the paragraph before Lemma 1. Actually, the idea of running multiple instances of base algorithm is widely used in non-stationary bandit algorithm since it was proposed in AdSwitch (Auer et al. (2019)). Lastly, the regret calculation is significantly different. As mentioned at the end of Section 5.2, we introduce a new partitioning of the horizon into near-stationary intervals to improve the bound.
>
> Lastly, we have added some motivating examples with references in the introduction and incorporated the explanation for assumption 3 to the revised version. Please check the global response above for all modifications in the revised version.
>
> ```
> Chen-Yu Wei and Haipeng Luo. Non-stationary reinforcement learning without prior knowledge: an optimal black-box approach. 2021.
> Peter Auer, Pratik Gajane, and Ronald Ortner. Adaptively tracking the best bandit arm with an unknown number of distribution changes. 2019.
> ```

---

> > ### Author Response · Authors · 2023-11-21
> > **Sincerely Looking Forward to Further Feedbacks**
> >
> > Dear Reviewer,
> >
> > Thank you for your time and efforts in reviewing our work. We have provided detailed clarification to address the issues raised in your comments. If our response has addressed your concerns, we would be grateful if you could re-evaluate our work.
> >
> > If you have any additional questions or comments, we would be happy to have further discussions.
> >
> > Thanks,
> >
> > The authors

---

### Author Response · Authors · 2023-11-15
**Summary of Changes in the Revised Version**

We thank all reviewers for your careful review and constructive suggestions! Here, we summarize the major changes we made to the revised paper.
- We added more motivating examples to the Introduction with references. We also added explicit definition of black-box algorithms in this section.
- We fixed the typo in Definition 4. $\Delta$ takes the max over state and actions instead of the sum. This is in accordance with prior works on non-stationary RL, such as Wei & Luo (2021).
- We added comments on what would happen if the environment degenerate to the stationary setting.
- We added explanation on why Assumption 3 is common. In particular, we prove that Assumption 1 in Wei & Luo (2021) implies Assumption 3. Details can be found in the proof of Proposition 3.
- We fixed the typo in Proposition 3. $c_2^\Delta=O(H^2)$ instead of $c_2^\Delta=O(H)$.
- We changed all 'Protocol' to 'Algorithm' for coherence.
- We fixed several word/grammar typos.
```
Chen-Yu Wei and Haipeng Luo. Non-stationary reinforcement learning without prior knowledge: an optimal black-box approach. 2021.
```

---

### Meta-Review · Area_Chair_MVr9 · 2023-12-11

**Metareview:**

The paper focuses on bounding the regret in Markov games with changing transition model and rewards under bandit feedback. The paper provides a reduction from stationary environments, i.e., given an algorithm that has regret guarantees for fixed transition model/rewards, an algorithm is suggested that achieves regret sublinear in T if the nonstationarity measure (path lengths of consecutive transition matrices and rewards or number of changes) is sublinear. The reviewers believe that the paper has merits and is slightly above the bar, despite some writeup weaknesses. We advise the reviewers to improve the writeup based on the reviewers questions and the feedback.

**Justification For Why Not Higher Score:**

The regret analysis probably is not tight with respect to Delta and the result is based heavily on previous papers. The writeup needs improvement.

**Justification For Why Not Lower Score:**

Interesting results and bandit feedback is challenging.

---

### Decision · Program_Chairs · 2024-01-16

Accept (poster)